# SharpZO: Hybrid Sharpness-Aware Vision Language Model Prompt Tuning via Forward-Only Passes

Yifan Yang[1], Zhen Zhang[1], Rupak Vignesh Swaminathan[2], Jing Liu[2], Nathan Susanj[2],
Zheng Zhang[1]

[1]University of California, Santa Barbara
[2]Amazon AGI

{yifanyang, zhen_zhang}@ucsb.edu    {swarupak, jlmk, nsusanj}@amazon.com
zhengzhang@ece.ucsb.edu
Project Page: https://yifan-yang.net/sharpzo.github.io/

## Abstract

Fine-tuning vision language models (VLMs) has achieved remarkable performance across various downstream tasks; yet, it requires access to model gradients through backpropagation (BP), making them unsuitable for memory-constrained, inference-only edge devices. To address this limitation, previous work has explored various BP-free fine-tuning methods. However, these approaches often rely on high-variance evolutionary strategies (ES) or zeroth-order (ZO) optimization, and often fail to achieve satisfactory performance. In this paper, we propose a hybrid Sharpness-aware Zeroth-order optimization (SharpZO) approach, specifically designed to enhance the performance of ZO VLM fine-tuning via a sharpness-aware warm-up training. SharpZO features a two-stage optimization process: a sharpness-aware ES stage that globally explores and smooths the loss landscape to construct a strong initialization, followed by a fine-grained local search via sparse ZO optimization. The entire optimization relies solely on forward passes. Detailed theoretical analysis and extensive experiments on CLIP models demonstrate that SharpZO significantly improves accuracy and convergence speed, achieving up to 7% average gain over state-of-the-art forward-only methods.

## 1 Introduction

In recent years, fine-tuning vision-language models (VLMs) has achieved remarkable performance across a wide range of downstream tasks, including image classification [54, 55], object detection [11, 52], and image segmentation [45, 24]. Among these models, one of the most prominent is CLIP [34], which has attracted significant attention for its powerful zero-shot recognition capabilities. To further improve the performance of VLMs in downstream tasks, previous work has explored the use of efficient, trainable prompt parameters [55, 49, 43] for the prompt tuning of VLMs. However, these prompt-tuning techniques are heavily dependent on the availability of a backward computation engine, which is typically unavailable on memory-constrained edge devices used in Internet-of-Things (IoT) applications [40] or wearable technologies [8].

To address these limitations, recent studies have explored fine-tuning VLMs in backpropagation-free settings [38, 49, 43]. These approaches optimize trainable prompts by leveraging high-variance black-box optimization techniques such as Evolutionary Strategies (ES) [17, 2] and Zeroth-Order (ZO) optimization [31, 32, 53] as alternatives to the first-order (FO) methods used in white-box scenarios. For instance, [49] employ ES to update prompt parameters by evaluating sampled prompts through forward passes only, thereby eliminating the need for memory-expensive back propagation. More recently, ZO stochastic gradient descent (SGD) [14] methods have been adapted to VLM

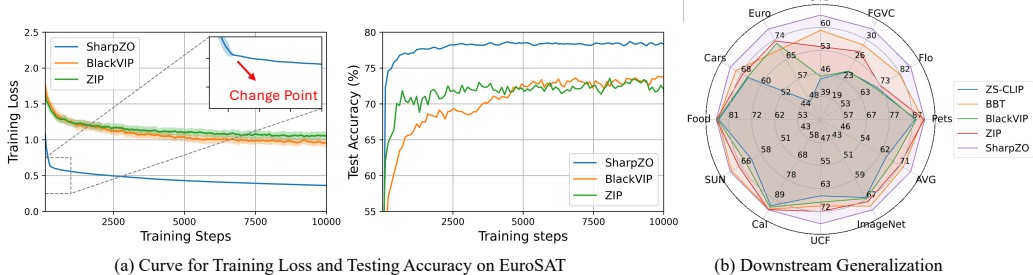

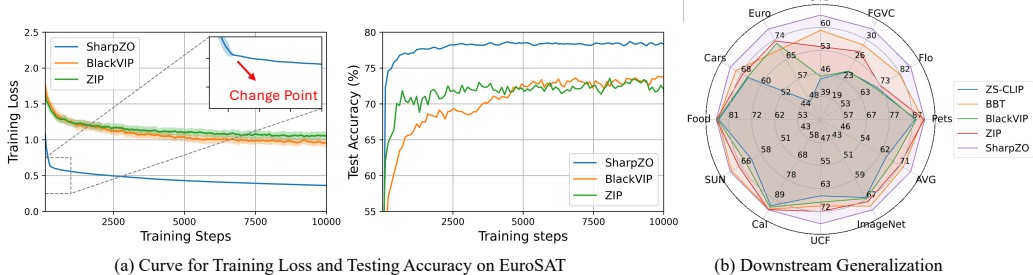

(a) Curve for Training Loss and Testing Accuracy on EuroSAT  (b) Downstream Generalization

Figure 1: (a) Comparison between SharpZO and other ZO prompt-tuning baselines.SharpZO demonstrates significantly lower variance than other ZO-based baselines like ZIP [32] and BlackVIP [31]. (b) Fine-tuned performance across all 11 tasks tested compared with ZIP and BlackVIP and BBT [39]. All experiments are conducted using the CLIP model with a ViT-B/16 backbone.

fine-tuning in the work of BlackVIP and ZIP [31, 32]. By approximating gradients with just two forward evaluations, these ZO approaches avoid the high computational cost and instability of ES, yet still match its performance while requiring substantially fewer model queries [32].

However, existing ZO-based VLM fine-tuning methods remain substantially inferior to backpropagation-based training. Their high variance and inherently local search dynamics make them prone to premature convergence. Previous work has attempted to improve the performance of ZO optimization by reducing the problem dimensionality through pruning [15, 26, 50] and low-rank decomposition [46, 32] of the trainable parameters. However, in widely adopted prompt-tuning settings, such parameter reduction offers limited benefit, as the number of trainable parameters is already inherently small in original trainable prompt.

In contrast to prior work that reduces the variance of ZO optimization by limiting the number of trainable parameters, our approach introduces a new perspective that focuses on initialization and the sharpness of the loss landscape. Specifically, we propose SharpZO, a hybrid Sharpness-Aware Zeroth-Order optimization method that employs a two-stage framework to significantly reduce the variance of ZO gradient estimation and improve the performance of ZO-based VLMs prompt tuning.

The first stage performs warm-up training using a sharpness-aware Covariance Matrix Adaptation Evolution Strategy (CMA-ES), which provides both a smoother loss landscape and a strong initialization for the second stage. Unlike gradient-based methods that follow local descent directions, CMA-ES enables effective global exploration by adaptively shaping the search distribution based on past evaluations [29]. Moreover, incorporating sharpness not only improves model generalization but also improves the accuracy of the randomized gradient estimators used in stage 2 ZO training, which is unbiased only with respect to a smoothed version of the objective function [14].

In the second stage, we perform fine-grained local optimization using a sparse Zeroth-Order Stochastic Gradient Descent (ZO-SGD) method. To further reduce gradient estimation variance, we introduce a novel *Z-pruning* technique specifically designed for noisy ZO settings, effectively reducing the dimensionality of the search space. Unlike conventional magnitude-based pruning used in previous sparse ZO method [15, 26], Z-pruning leverages gradient information to capture the influence of model non-linearity and applies Z-score-based normalization [12] to suppress outlier gradient estimates.

As shown in Figure 1, our method converges faster with significantly lower variance compared with other ZO prompt-tuning baselines, achieving up to a 7% average improvement in accuracy. Our main contributions are summarized as follows:

- We propose SharpZO, a novel hybrid sharpness-aware optimizer that fine-tunes VLMs using only forward passes. To our knowledge, this is the first ZO method that improves performance considering the sharpness-aware initialization.
- In the first stage, we introduce a sharpness-aware CMA-ES that enhances generalization and reduces second stage ZO gradient estimation variance by smoothing the loss landscape.
- In the second stage, we develop a sparse ZO fine-tuning method with a novel Z-pruning technique to suppress outliers in noisy gradient estimates.
- We validate SharpZO through extensive experiments and theoretical analysis, demonstrating superior performance over existing BP-free baselines.

## 2 Background

### 2.1 Coordinate-wise Gradient Estimation or Randomized Gradient Estimation?

Mainstream ZO gradient estimation methods can be broadly classified into two categories: Coordinate-wise Gradient Estimation (CGE) and Randomized Gradient Estimation (RGE). In our SharpZO framework, we employ CGE to compute sharpness-related terms in stage 1 and pruning metrics in stage 2, while RGE is used to update parameters during the second-stage ZO-SGD optimization. Below, we provide background on both approaches and highlight their key differences in terms of estimation variance and computational efficiency.

Given a VLM with trainable prompt vector $w \in \mathbb{R}^d$, we define the training cross-entropy loss as $\mathcal{L}(w)$. ZO estimated gradients $\nabla_w \mathcal{L}(w)$ are estimated via forward differences between function evaluations, where the perturbation of the trainable parameters $w$ depends on whether the CGE or RGE method is used, which gives:

$$\textbf{(RGE)} \ \hat{\nabla}\mathcal{L}(w) = \frac{1}{q}\sum_{i=1}^{q}\left[\frac{\mathcal{L}(w+\mu u_i) - \mathcal{L}(w - \mu u_i)}{2\mu}u_i\right]; \ \textbf{(CGE)} \ \hat{\nabla}\mathcal{L}(w) = \sum_{i=1}^{d}\left[\frac{\mathcal{L}(w+\mu e_i) - \mathcal{L}(w - \mu e_i)}{2\mu}e_i\right]. \ (1)$$

Here, $\mu > 0$ is a smooth parameter, $u_i \in \mathbb{R}^d$ denotes a randomized perturbation vector perturbing all parameters at the same time in RGE, and $e_i = [0, 0, \cdots, 1, \cdots, 0]^T$ represents the $i$-th standard basis vector, which is used to compute a finite-difference approximation of $\mathcal{L}(w)$ along a single coordinate in CGE. To reduce the variance of the RGE gradients, it is common to average the gradients estimated over $q$ different randomized perturbations, where $q$ is called query numbers. In contrast, the CGE method approximates the gradient by perturbing individual coordinates and estimating the directional derivative along each axis using finite differences.

Unlike FO methods that compute exact gradients $\nabla\mathcal{L}(w)$ via BP, RGE methods estimate gradients in a biased manner toward the exact gradients, which instead provides an unbiased estimate of the gradient of a smoothed version of the objective, defined as $\mathcal{L}_\mu(w) = \mathbb{E}_u[\mathcal{L}(w + \mu u)]$. In contrast, CGE estimates directional derivatives along individual coordinates without applying such smoothing, resulting in greater sensitivity to sharp changes in the loss landscape.

**Difference between RGE and CGE:** We compare RGE and CGE primarily in terms of query complexity and accuracy. The number of function queries differs significantly between the two methods. Given the dimension of trainable parameter as $d$ and the number of RGE query as $q$, RGE requires $\mathcal{O}(q)$ queries, whereas CGE incurs a higher cost of $\mathcal{O}(d)$ queries. Clearly, RGE offers much lower query complexity, especially when $q = 1 \ll d$ in our case. Despite its higher query complexity, CGE achieves superior accuracy compared to RGE, as it directly approximates the true gradient without introducing smoothed objective function [5].

### 2.2 Covariance Matrix Adaptation Evolution Strategy (CMA-ES)

Similar as ZO method, CMA-ES is another type of derivative-free optimizer for continuous, black-box functions [18]. CMA-ES achieves truly global search by maintaining and adapting a full covariance matrix, which captures variable interactions and shapes an anisotropic search distribution; sampling a population each generation then naturally balances broad exploration (through a larger step-size) with focused exploitation (via covariance updates). In contrast, zeroth-order methods rely on local gradient approximations at a single point and random perturbation directions.

At iteration $t$, CMA-ES maintains parameter $\theta_t$, $\sigma_t$ and $C_t$, where $\theta_t$ is the search-distribution mean, $\sigma_t$ is the global step size, and $C_t$ the covariance matrix. To update these parameter at each iteration, a population of $S$ candidates is drawn as

$$w_t^i \sim \theta_t + \sigma_t \mathcal{N}(0, C_t), \quad i = 1, \ldots, S,$$

and their fitness is evaluated via the black-box loss $\mathcal{L}(w_t^i)$.

After evaluating the population, the parameters are updated in three steps. First, the mean of the distribution $\theta_t$ is shifted toward regions with lower loss $\mathcal{L}(w)$ by taking a weighted combination of the top-performing candidates $w^i$. This moves the sampling center toward promising areas in the search space while maintaining stochasticity. Second, the covariance matrix $C_t$ is adapted to capture both the overall spread and the correlations among the selected samples. Finally, the global step size

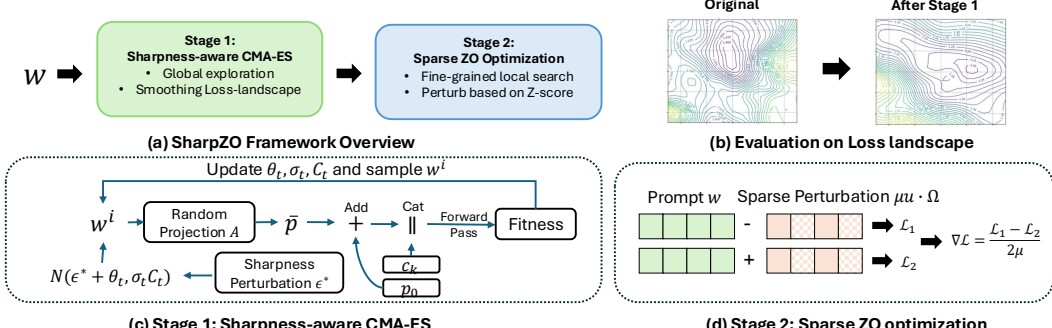

Figure 2: Overview of the SharpZO method. (a) The overall training pipeline of SharpZO, consisting of a two-stage optimization process. (b) Visualization of the smoothed loss landscape after Stage 1 sharpness-aware CMA-ES optimization. (c) Training dynamics of the sharpness-aware CMA-ES method. (d) RGE-based gradient estimation during sparse ZO training in Stage 2.

$\sigma_t$ is adjusted through a step-size adaptation mechanism, which regulates the overall exploration scale based on the recent success of the search. For detailed derivations and algorithmic formulations, please refer to [18].

## 2.3 Sharpness-aware Optimization

Sharpness-Aware Minimization (SAM) [13] was originally proposed to improve generalization by smoothing the loss landscape and encouraging the optimization process to converge to flat minima. The key idea is to minimize the worst-case loss within a neighborhood around the current parameters $\boldsymbol{w}$ by introducing an analytic approximation of the worst-case perturbation $\boldsymbol{\epsilon}^*$ within a radius $\rho$. Specifically, the SAM objective is formulated as:

$$\min_{\boldsymbol{w}} \mathcal{L}_\rho^{\text{SAM}}(\boldsymbol{w}), \quad \text{where} \quad \mathcal{L}_\rho^{\text{SAM}}(\boldsymbol{w}) = \max_{\|\boldsymbol{\epsilon}\|_2 \leq \rho} \mathcal{L}(\boldsymbol{w} + \boldsymbol{\epsilon}).$$

Following the objective function, the gradient approximation for SAM after dropping the second-order terms is given as $\nabla_{\boldsymbol{w}} \mathcal{L}(\boldsymbol{w}')|_{\boldsymbol{w}' + \boldsymbol{\epsilon}^*}$ by computing the worst-case perturbation $\boldsymbol{\epsilon}^*$. To estimate $\boldsymbol{\epsilon}^*$, SAM approximates the inner maximization using a FO Taylor expansion of the loss function. This yields the following analytic solution:

$$\boldsymbol{\epsilon}^* \approx \arg \max_{\|\boldsymbol{\epsilon}\|_2 \leq \rho} \left( \mathcal{L}(\boldsymbol{w}) + \boldsymbol{\epsilon}^\top \nabla \mathcal{L}(\boldsymbol{w}) \right) = \rho \cdot \nabla \mathcal{L}(\boldsymbol{w}) / \|\nabla \mathcal{L}(\boldsymbol{w})\|_2. \tag{2}$$

Unlike previous works that apply SAM to replace standard gradient descent either throughout the entire training process [13] or during the final few epochs [56], we, for the first time, investigate the effectiveness of incorporating sharpness information into CMA-ES as an early-stage warm-up strategy to enhance the performance of ZO fine-tuning.

## 3 The SharpZO Method

In this section, we detail the SharpZO method, which is designed to fine-tune VLMs using only forward passes. As illustrated in Fig. 2 (a), our approach consists of two main stages: a sharpness-aware CMA-ES stage and a sparse ZO fine-grained search stage. We demonstrate that performing a sharpness-aware global search in the early steps of training significantly enhances the performance of ZO optimization—both in terms of convergence speed and final accuracy. In the following, we first summarize the problem setup of this work and then describe each stage of our method in detail.

We consider a black-box VLM with loss function $\mathcal{L}(\boldsymbol{w})$ and a dataset $\mathcal{D} = \{(x_n, y_n)\}_{n=1}^N$ with a total of $N$ samples and $K$ classes. For models like CLIP, it is necessary to construct a text prompt for each class. Specifically, for a given class $k \in \{1, \dots, K\}$ and the model hidden dimension $m$, the class-specific prompt $\boldsymbol{p}^k$ is defined by concatenating a predefined initial text embedding $\boldsymbol{p}_0 \in \mathbb{R}^m$ (e.g., "a photo of a") with the class label embedding $\boldsymbol{c}^k$, yielding $\boldsymbol{p}^k = [\boldsymbol{p}_0, \boldsymbol{c}^k]$. To make the prompt $\boldsymbol{p}^k$ trainable, we introduce a parameter $\bar{\boldsymbol{p}}_t$ that modifies the initial embedding $\boldsymbol{p}_0$, where $t \in [1, T]$

is the training steps. This parameter is obtained via a random projection from a low-dimensional trainable matrix $\boldsymbol{w}_t \in \mathbb{R}^d$, where $d \ll m$ is the latent dimension. Then, a fixed randomized projection matrix $\boldsymbol{A} \in \mathbb{R}^{m \times d}$ is used to project $\boldsymbol{w}$ into the embedding space, producing $\bar{\boldsymbol{w}}_t = \boldsymbol{w}_t \boldsymbol{A}^\top \in \mathbb{R}^m$ to matching the shape of the original initialized prompt $\boldsymbol{p}_0$. Thus, the overall training objective becomes:

$$\min_{\boldsymbol{w}} \frac{1}{N} \sum_{n=1}^{N} \mathcal{L}([\boldsymbol{p}_0 + \bar{\boldsymbol{p}}_t, c_k], (x_n, y_n)), \quad \text{where} \quad \bar{\boldsymbol{p}} = \boldsymbol{w}_t \boldsymbol{A}^\top$$

Next, we provide a detailed introduction to the two stages of the SharpZO method separately.

### 3.1 Stage 1: Sharpness-Aware CMA-ES Method

In this section, we first summarize the traditional CMA-ES method and then propose our sharpness-aware CMA-ES optimization for a warm-up training.

In the traditional CMA-ES method, given the population size $S$, the optimizer generates a population of candidate solutions $\boldsymbol{w}_t^i, i \in [1, S]$, where $\boldsymbol{w}_t^i$ is obtained by sampling from the current multivariate normal distribution $\boldsymbol{w}_t^i \sim \boldsymbol{\theta}_t + \sigma_t \mathcal{N}(0, \boldsymbol{C}_t)$. Here, $\boldsymbol{\theta}_t$ is the weighted mean of the distribution, $\sigma$ is the step size, $S$ is the population size and $\boldsymbol{C}$ is the covariance matrix capturing the shape and scale of the search distribution. After evaluating the loss (fitness) $\mathcal{L}(\boldsymbol{w}_t^i, \mathcal{D})$ of these samples by forwarding the training samples $\mathcal{D}$ along with the trainable prompt $\boldsymbol{w}_t^i, i \in [1, S]$, the parameters $\boldsymbol{\theta}_t, \sigma_t$, and $\boldsymbol{C}_t$ are updated accordingly [17].

Different from the previous CMA-ES method, we propose a new sharpness-aware CMA-ES method to smooth the loss landscape during the stage 1 warm-up training, which help to reduce the stage 2 gradient estimation accuracy. Specifically, we add the worst-case perturbation $\boldsymbol{\epsilon}^*$ during the sampling of CMA-ES method, where $\boldsymbol{\epsilon}^*$ is computed within a local Euclidean ball based on eq. (2) and gives:

$$\boldsymbol{w}^i \sim \boldsymbol{\epsilon}_t^* + \boldsymbol{\theta}_t + \delta_t \mathcal{N}(0, \boldsymbol{C}_t), \quad i \in [1, S], \quad \boldsymbol{\epsilon}_t^* = \rho \frac{\nabla \mathcal{L}(\boldsymbol{\theta}_t)}{\|\nabla \mathcal{L}(\boldsymbol{\theta}_t)\|_2}. \tag{3}$$

The effectiveness of this modification can be explained through the Taylor expansion of the Monte Carlo estimation for the loss $\mathbb{E}[\mathcal{L}(\boldsymbol{\theta}_t + \boldsymbol{\epsilon}^* + o)]$ used in the sharpness-aware CMA-ES optimizer, given $o \sim \mathcal{N}(0, \delta_t^2 \boldsymbol{C}_t)$. Given the sampling strategy in eq. (3) and omitting the first-order term by the fact $\mathbb{E}[o] = 0$, the expected fitness can be approximated as:

$$\mathbb{E}[\mathcal{L}(\boldsymbol{\theta}_t + \boldsymbol{\epsilon}^* + o)] \approx \mathcal{L}(\boldsymbol{\theta}_t + \boldsymbol{\epsilon}^*) + \frac{1}{2}\mathbb{E}[o^\top \nabla^2 \mathcal{L}(\boldsymbol{\theta}_t + \boldsymbol{\epsilon}^*)o], \quad o \sim \mathcal{N}(0, \delta_t^2 \boldsymbol{C}_t).$$

As observed, in addition to optimizing the same term $\mathcal{L}(\boldsymbol{\theta}_t + \boldsymbol{\epsilon}^*)$ as gradient-based SAM methods, the effectiveness of the sharpness-aware CMA-ES approach is achieved with additional higher order term involving stochastic adaptation of the covariance matrix, which introduces an additional mechanism to explore and down-weight high-curvature directions.

Another challenge in applying sharpness-aware CMA-ES lies in gradient estimation. Specifically, we cannot directly access the gradient information needed to compute $\boldsymbol{\epsilon}^*$ due to the absence of backpropagation. Unlike prior work [51, 48], which relies on RGE-based gradient estimation with a large query budget $q$, we adopt CGE for this purpose. CGE provides an unbiased estimate of the gradient along each coordinate, making it more suitable for computing the sharpness-aware perturbation term, as discussed in Section 2.1. The detailed formulation for estimating $\nabla \mathcal{L}(\boldsymbol{w})$ using CGE is provided in eq. (1).

### 3.2 Stage 2: Fine-grained ZO Method

After obtaining a strong initialization and a smoothed loss landscape through the first-stage global search, we proceed to optimize the parameters toward the global optimum using a ZO-SGD method. In this stage, we perform fine-grained and efficient local search guided by RGE-estimated gradients. To further reduce the variance, we propose a new Z-pruning metrics with pruning mask $\Omega$ to reduce the effective dimension during gradient estimation.

Unlike prior sparse ZO methods that rely solely on magnitude-based pruning metrics, we introduce a novel pruning criterion tailored to the high-variance nature of ZO-estimated gradients. In contrast to first-order sparse training approaches, which apply the pruning mask $\Omega$ directly to model weights,

we apply the mask to the perturbation vector $\boldsymbol{u}$. This strategy effectively reduces the dimensionality during ZO perturbation and results in the following gradient estimator:

$$\hat{\nabla}_{\boldsymbol{w}}\mathcal{L}(\boldsymbol{w}) = \frac{\mathcal{L}(\boldsymbol{w} + \mu \cdot \Omega\boldsymbol{u}) - \mathcal{L}(\boldsymbol{w} - \mu \cdot \Omega\boldsymbol{u})}{2\mu}\boldsymbol{u} \tag{4}$$

We note that a query number of $q = 1$, as defined in Eq. (1), is sufficient for stable training of our SharpZO method when preceded by the Stage 1 warm-up. This is significantly more efficient than prior works such as ZIP [32] and BLACKVIP [31], which still exhibit high training variance even with a query number of $q = 5$, as shown in Fig. 1. Next, we introduce how we construct the Z-pruning based pruning mask $\Omega$.

**Z-pruning Metrics:** The Z-pruning metric is designed to minimize the loss degradation introduced by pruning, by considering the sensitivity of each parameter. Specifically, considering $\delta\boldsymbol{w}$ reflect the change of weights during pruning, the difference in loss between the dense and pruned models can be approximated via a first-order Taylor expansion:

$$\mathcal{L}(\boldsymbol{w} + \delta\boldsymbol{w}) - \mathcal{L}(\boldsymbol{w}) = \nabla\mathcal{L}(\boldsymbol{w})^\top \delta\boldsymbol{w} + \frac{1}{2}\delta\boldsymbol{w}^\top H \delta\boldsymbol{w} + \mathcal{O}(\|\delta\boldsymbol{w}\|^3),$$

where $H$ denotes the Hessian of the loss with respect to the parameters. Here, we can approximate the Hessian $H$ using a Fisher matrix as $H \approx \mathbb{E}_{x \sim \mathcal{D}}[\nabla\mathcal{L}(\boldsymbol{w}; x)^2]$ and estimating gradients $\nabla\mathcal{L}(\boldsymbol{w}; x)$ using the CGE method described in eq. (1) as $\hat{\nabla}\mathcal{L}(\boldsymbol{w}; x)$. By considering the second-order term, we can obtain the Z-pruning metrics as:

$$\Omega = |\boldsymbol{w}|^2 \cdot z(\mathbb{E}_{x \sim \mathcal{D}}[\hat{\nabla}\mathcal{L}(\boldsymbol{w}, x)^2]), \quad \text{where } \hat{\nabla}\mathcal{L}(\boldsymbol{w}) = \sum_{i=1}^d \left[\frac{\mathcal{L}(\boldsymbol{w} + \mu\boldsymbol{e}_i) - \mathcal{L}(\boldsymbol{w} - \mu\boldsymbol{e}_i)}{2\mu}\boldsymbol{e}_i\right], \tag{5}$$

where $z(\boldsymbol{g}^2) = (\boldsymbol{g}^2 - \mu_g)/\sigma_g$ denote the Z-score normalization given $\mu_g$ and $\sigma_g$ as the mean and standard deviation of the gradient vector $\boldsymbol{g} = \hat{\nabla}\mathcal{L}$. The Z-score standardizes the gradient magnitudes to mitigate the scale mismatch between trainable parameters and their gradients—a mismatch that is especially pronounced in ZO settings. We consider adapting the second-order term as the pruning score based on the practice of previous pruning work [37, 47]. In Section 5.4.2, we demonstrate the effectiveness of the Z-pruning method compared to dense and magnitude-based pruned ZO training.

### 3.3 Algorithms

We present the SharpZO algorithm in Algorithm 1. In practice, we first execute Stage 1 for a total of $T_c$ steps, followed by Stage 2 until the total training budget of $T$ steps is reached. The transition point $T_c$ is determined automatically using a strategy inspired by early stopping, based on the observed change in validation accuracy. Typically, $T_c \ll T$ and is typically reached within 100 steps. As a result, although Stage 1 is more computationally expensive per step, the overall training efficiency remains high. During Stage 2, we update the pruning mask every $K$ steps to balance computational cost and adaptability to the evolving optimization landscape. The learning rate for ZO optimization in Stage 2 is denoted by $\eta$.

## 4 Theoretical Guarantee

In this section, we give a quick proof to show the convergence rate of SharpZO method. By comparing the SharpZO convergence rate with the baseline ZO-SGD rate given in MeZO [28], we highlight why our hybrid smoothness-aware setup can significantly help to improve the performance of VLMs fine-tuning. To align our analysis with VLMs/LLMs fine-tuning, we consider a non-convex optimization setup and the proof assume the loss landscape follows the Polyak-Lojasiewicz (PL) inequality, which has been widely considered in other ZO fine-tuning papers [28]. First, we list the following assumptions for our analysis include the PL-inequality we just mentioned:

**A1 (PL Inequality):** The loss function $\ell$ satisfies the Polyak–Łojasiewicz (PL) condition. That is, there exists a constant $\mu > 0$ such that for all $\boldsymbol{w} \in \mathbb{R}^d$, we have $\frac{1}{2}\|\nabla\mathcal{L}(\boldsymbol{w})\|^2 \geq \mu\left(\mathcal{L}(\boldsymbol{w}) - \mathcal{L}^*\right)$, where $\mathcal{L}^*$ denotes the global minimum of the loss function.

**A2 (Lipschitz smoothness):** The loss function $\mathcal{L}$ has an $L$-Lipschitz continuous gradient. That is, there exists a constant $L > 0$ such that for all $\boldsymbol{w}_i, \boldsymbol{w}_j \in \mathbb{R}^d$, we have $\|\nabla\mathcal{L}(\boldsymbol{w}_i) - \nabla\mathcal{L}(\boldsymbol{w}_j)\| \leq L\|\boldsymbol{w}_i - \boldsymbol{w}_j\|$.

**Algorithm 1** SharpZO: Hybrid Sharpness-Aware Zeroth-Order Optimization

---

**Require:** Initial prompt parameters $\boldsymbol{w}_0$, total steps $T$, transition step $T_c$, pruning interval $K$
1: **for** $t = 1$ to $T$ **do**
2:    **if** $t < T_c$ **then**                                                   ▷ Stage 1: Sharpness-aware CMA-ES
3:       Sample candidate solutions $\boldsymbol{w}_t^i$ using eq. (3)
4:       Evaluate fitness $\mathcal{L}(\boldsymbol{w}_t^i)$ for each candidate
5:       Update CMA-ES parameters $\boldsymbol{\theta}_t$, $\sigma_t$, and $\boldsymbol{C}_t$ based on fitness values
6:    **else**                                                         ▷ Stage 2: Sparse ZO Optimization
7:       **if** $t = T_c$ or $(t - T_c) \bmod K = 0$ **then**
8:          Update pruning mask $\Omega$ using eq. (5)
9:       **end if**
10:      Estimate gradient $\hat{\nabla}\mathcal{L}(\boldsymbol{w}_t)$ using the ZO oracle (eq. 4)
11:      $\boldsymbol{w}_{t+1} \leftarrow \boldsymbol{w}_t - \eta \cdot \hat{\nabla}\mathcal{L}(\boldsymbol{w}_t)$
12:    **end if**
13: **end for**
14: **return** Fine-tuned prompt vector $\boldsymbol{w}_T$

---

**Theorem 1.** *Under assumptions A1 and A2, suppose the SharpZO algorithm first performs $T_c$ steps of global optimization using CMA-ES and then switches to zeroth-order gradient-based optimization until convergence. The convergence rate of SharpZO method can be give by:*

$$t \approx T_c + \mathcal{O}\left(\frac{1}{\eta\mu}\log\left(\frac{L^3\eta^2\rho^2}{\epsilon}\right)\right) \tag{6}$$

*where $\epsilon$ is given by assuming $\mathcal{L}(\boldsymbol{w}_t) - \mathcal{L}^* \leq \epsilon$, $\eta$ is the learning rate of ZO-SGD optimizer in stage 2 and $L$ is the smoothness factor. Eq. (6) is obtained by ignoring the lower order terms for clarity.*

*Proof.* Details of the proof can be found in Appendix D. ☐

Compared to the naive ZO-SGD convergence rate presented in [28], the SharpZO method leverages a sharpness-aware initialization strategy that yields a lower starting point for the second-stage ZO training, specifically of the form $(1 - \mu(1 - 2L\sigma^2))^{T_c}\Delta_0$ in eq. (13) in Appendix D. Since we use a relative large step size $\sigma$ during the first stage training, we can observe the original error gap $\Delta_0$ is linearly decreasing with high scaling factor. Moreover, as we can observe from the sharpness term in eq. (6), the integration of sharpness-aware optimization effectively clips the L-smoothness constant $L$ with the sharpness parameter $\rho$, thereby reducing the effective sharpness in the ZO training phase.

## 5 Experiments

In this section, we present experimental results to evaluate the performance of the proposed SharpZO method across a variety of downstream tasks using CLIP models with different architectures. Specifically, we compare the proposed SharpZO method with zero-shot (ZS) inference and other BP-free baselines like BBT [49], BlackVIP [31], and ZIP [32] (Detailed descriptions for tasks and baslines method can be found in Appendix B). Our results demonstrate that SharpZO not only achieves superior accuracy but also improves efficiency, as measured by the time-to-test-accuracy (ToTA) metric [7]. Additionally, we provide a comprehensive ablation study to analyze the contributions of individual components in Section 5.4.1 and Section 5.4.2. Further implementation details and extended experimental results—including evaluations across various model architectures and hyperparameter choices, as well as comparisons with state-of-the-art prompt-tuning methods that involve backpropagation, such as CraFT [43] are provided in Appendix C.

**Training Detail:** For the VLM model, we utilize CLIP [34] with both ResNet [19] and ViT [10] backbones as the visual encoder, and Transformers [41] as the text encoder. The CLIP weights are initialized from the official pretrained checkpoints and remain frozen during training. The prompt generator use initial prompt with length of $4$, and hidden dimension $d = 512$. Parameters in $\boldsymbol{w}$ are initialized from a Gaussian distribution $\mathcal{N}(0, 0.02)$.

Here, we manually tuned the change points from stage 1 to stage 2 by choosing from a set of parameter between $100$ to $500$. However, we also tried to adapt an early stopping criterion to automatically

Table 1: Few-shot performance across 11 datasets using CLIP models with ResNet and ViT backbones, trained for 20K steps. * indicate results reported in prior works [31, 43, 32]. We additionally reproduce the ZIP results, as the original paper restricted the query budget to 5K. Bold values highlight the best performance, demonstrating the superiority of SharpZO over all BP-free baselines.

| Backbone | Methods | ImageNet | Pets | Flo | FGVC | DTD | Euro | Cars | Food | SUN | Cal | UCF | AVG |
|---|---|---|---|---|---|---|---|---|---|---|---|---|---|
| | ZS-CLIP* | 58.18 | 85.77 | 66.14 | 17.28 | 42.32 | 37.56 | 55.61 | 77.31 | 58.52 | 86.29 | 61.46 | 58.77 |
| | BBT* | 61.74 | 88.73 | 72.53 | 12.07 | 54.33 | 69.01 | 60.24 | 78.44 | 64.34 | 90.05 | 67.91 | 65.40 |
| RN50 | BlackVIP | 60.33 | 85.99 | 65.12 | 17.37 | 42.73 | 58.16 | 56.70 | 77.23 | 59.17 | 86.37 | 60.11 | 60.84 |
| | ZIP | 61.30 | **89.53** | 68.41 | 19.98 | 47.40 | 63.10 | 58.61 | 78.98 | 62.86 | 90.63 | 64.05 | 64.08 |
| | SharpZO | **63.29** | 89.51 | **79.50** | **23.97** | **60.58** | **80.77** | **60.58** | **79.28** | **66.17** | **91.24** | **72.43** | **69.76** |
| | ZS-CLIP* | 66.73 | 89.21 | 71.34 | 24.72 | 44.39 | 47.60 | 65.32 | 86.06 | 62.50 | 92.94 | 66.75 | 65.23 |
| | BBT* | 70.15 | 92.70 | 82.41 | 29.49 | 59.26 | 70.48 | 70.19 | 86.42 | 70.33 | 94.75 | 70.48 | 72.42 |
| ViT-B/16 | BlackVIP* | 67.10 | 89.70 | 70.60 | 24.78 | 45.20 | 73.10 | 65.60 | 86.60 | 64.70 | 93.70 | 69.10 | 68.20 |
| | ZIP (Offical)* | 66.20 | 94.00 | 70.40 | 26.80 | 47.80 | 64.60 | 71.09 | 86.40 | 63.30 | 94.00 | 69.80 | 70.57 |
| | ZIP (Rep) | 68.35 | 93.18 | 73.00 | 28.32 | 54.26 | 74.19 | 67.58 | 87.01 | 67.43 | 94.97 | 72.51 | 70.98 |
| | SharpZO | **71.60** | **94.06** | **88.02** | **32.34** | **63.95** | **79.42** | **72.50** | **87.13** | **70.86** | **95.09** | **77.08** | **75.64** |

Table 2: Comparison of robustness to distribution shift between SharpZO and other baselines. The best results among BP-free methods are highlighted in bold.

| Method | ResNet-50 | | | | | | ViT-B/16 | | | | | |
|---|---|---|---|---|---|---|---|---|---|---|---|---|
| | ImageNet | -V2 | -Sketch | -A | -R | Avg | ImageNet | -V2 | -Sketch | -A | -R | Avg |
| ZS-CLIP | 58.2 | 51.3 | 33.3 | 21.7 | 56.0 | 40.6 | 66.7 | 60.8 | 46.2 | 47.8 | 74.0 | 57.2 |
| CoOp | 63.3 | 55.4 | 34.7 | 23.1 | 56.6 | 42.4 | 71.7 | 64.6 | 47.9 | 49.9 | 75.1 | 59.4 |
| BBT | 61.7 | 54.0 | 33.9 | 23.2 | 58.3 | 42.4 | 70.2 | 63.0 | **47.9** | 49.5 | 76.1 | 59.1 |
| BlackVIP | 60.2 | 52.3 | 33.3 | 21.5 | 57.7 | 41.2 | 65.5 | 59.2 | 44.6 | 42.5 | 73.1 | 54.9 |
| ZIP | 61.3 | 53.7 | 33.7 | 23.9 | 57.6 | 42.2 | 68.4 | 59.7 | 45.5 | 47.1 | 75.2 | 56.9 |
| SharpZO | **63.3** | **54.8** | **35.2** | **24.5** | **58.7** | **43.3** | **71.6** | **63.8** | 45.0 | **50.3** | **76.6** | **58.9** |

decide this point: the algorithm switches to stage 2 if the validation accuracy does not improve by more than 0.01 over the best recorded accuracy for 10 consecutive steps, which can achieve similar results. Detail hyper-parameter setup for SharpZO method on various tasks can be found in Table. 8 in Appendix C.2. All experiments use a 16-shot setup unless otherwise specified.

## 5.1 Results on Few-Shot Classification

We first compare our SharpZO method with SOTA BP-free prompt-tuning baselines across 11 downstream tasks. To explore the effect of different model architectures, we evaluate all methods using CLIP models with both ResNet-50 and ViT-B/16 viusal encoder backbones. The results are summarized in Table 1. Based on these results, we draw the following conclusions:

**SharpZO significantly outperforms all other BP-free methods.** As shown in Table 1, SharpZO consistently surpasses other ZO prompt tuning approaches in terms of classification accuracy. Compared to the SOTA ZO prompt tuning method ZIP, SharpZO achieves an absolute average performance gain of **5%** and outperforms ZIP among **all** 11 tasks on the CLIP model with ViT-B/16 backbone. The performance of SharpZO is approaching first-order method like CoOp, which shows the potential of deploying ZO method in real-world application. These improvements are driven by the reduction of gradient estimation variance and bias with the sharpness-aware warm-up training.

**SharpZO performs robustly across diverse model architectures.** Unlike prior ZO prompt-tuning methods such as ZIP and BlackVIP—which often struggle to converge on certain tasks like Flowers102, EuroSAT, and UCF101 when using CLIP models with ResNet backbones—our proposed SharpZO method consistently delivers strong performance across a wide range of architectures and tasks. Additional results using architectures such as ResNet-101 and ViT-B/32, presented in Appendix C.1, further demonstrate the robustness of SharpZO to varying model backbones.

**SharpZO exhibits lower training variance.** As illustrated by the optimization curves in Fig. 1(a), SharpZO achieves markedly more stable training—its standard-deviation bands are substantially narrower than those of other ZO methods such as ZIP and BlackVIP.

## 5.2 Robustness to Distribution Shift

In this section, we further evaluate the robustness of the SharpZO method under distribution shifts. Results comparing SharpZO to other BP-free baselines are summarized in Table 2. Compared to the

Figure 3: Comparison between the naive CMA-ES with the Sharpness-aware (S-aware) CMA-ES method on EuroSAT dataset.

Figure 4: Comparison between the naive ZO optimization and sparse ZO optimization with various pruning metrics on EuroSAT dataset.

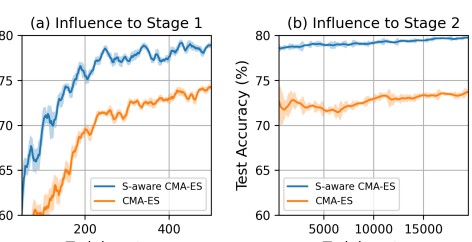

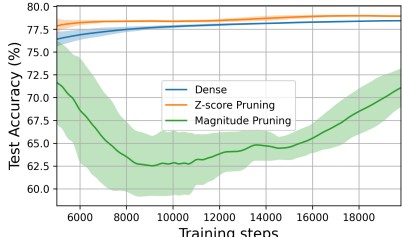

state-of-the-art ZO method ZIP, SharpZO achieves an absolute improvement of **2.0%** on ResNet-50 and **2.8%** on ViT-B/16 (averaged over all distribution shift benchmarks) for the ImageNet test accuracy. These findings highlight the strong out-of-distribution generalization ability of SharpZO under varying types of distribution shifts.

## 5.3 Time-to-test-accuracy Efficiency

In this section, we evaluate the training efficiency of the proposed SharpZO method. We focus on training time rather than memory usage, as all zeroth-order (ZO) baselines exhibit comparable memory consumption due to their forward-only nature. Specifically, we measure the wall-clock time required to reach a common evaluation accuracy threshold, following the protocol of [7]. The threshold is selected such that it is attainable by all baselines. The results are summarized in Table 3 and tested on single Nvidia A100-40G GPU.

| Methods | IN | Pets | DTD | Euro |
|---|---|---|---|---|
| BlackVIP | 172.6 | 714.8 | 132.0 | 201.5 |
| ZIP | 19.0 | 126.3 | 6.0 | 251.4 |
| SharpZO | **15.3** | **2.4** | **2.6** | **12.7** |

Table 3: Time-to-test accuracy comparison between different BP-free prompt tuning methods on multiple dataset. The time is recorded in minutes.

As shown in Table 3, the SharpZO method achieves faster convergence compared to other ZO prompt tuning baselines, which is consistent with our theoretical analysis in Section 4. Beyond the benefits of the proposed hybrid sharpness-aware optimization scheme, the improved training speed of SharpZO also stems from its lower per-step query count and significantly faster forward pass.

Specifically, unlike ZIP and BlackVIP, which require 10 queries per step to reduce training loss, SharpZO only requires 2 queries per step during Stage 2. Moreover, ZIP incurs substantial overhead due to its complex reconstruction process in forward pass, taking approximately 0.53 seconds per forward pass, whereas SharpZO requires only 0.0069 seconds. Consequently, the average per-step training time of SharpZO is markedly lower than other ZO baseline like ZIP.

## 5.4 Ablation on Components in Different Stages

### 5.4.1 Influence of Stage 1 Sharpness Aware Optimization

In this section, we aim to validate the effectiveness and illustrate the influence of our sharpness-aware CMA-ES method to both stage 1 and stage 2 training in SharpZO. Specifically, we compare the training curve between the naive CMA-ES method and our sharpness-aware (S-aware) CMA-ES method for both stage in Fig. 3 (a) and Fig. 3 (b), respectively. The experiments are conducted on the EuroSAT dataset using the CLIP model with a ViT-B/16 backbone. For the convenience of comparison, the transition point from Stage 1 to Stage 2 optimization is fixed at 500 steps.

As illustrated in Fig. 3 (a), the sharpness-aware CMA-ES method consistently achieves a faster convergence rate and superior final accuracy compared to the naive CMA-ES method in stage 1, which shows a better generalization ability. More importantly, the sharpness-aware training benefits the second-stage convergence of ZO optimization as observed from Fig. 3. The sharpness-aware warm-up training leads to a more stable Stage 2 training curve with reduced variance, which can be

attributed to the implicitly clipped smoothness factor introduced by the sharpness-aware updates, as discussed in the theoretical analysis in Section 4.

### 5.4.2 Effectiveness of Stage 2 Sparse ZO Optimization

We evaluate the effectiveness of our proposed Z-pruning strategy for sparse ZO optimization in Stage 2. Specifically, we compare it against magnitude-based pruning and dense training on the EuroSAT dataset using the CLIP ViT-B/16 backbone. Results are shown in Fig. 4.

Our findings show that sparse ZO training with Z-pruning reduces gradient variance and improves both accuracy and convergence speed compared to dense training. In contrast, magnitude-based pruning—commonly used in prior work [15, 26]—performs poorly in prompt-tuning due to the limited number of trainable parameters (512), which makes accurate pruning critical. Moreover, magnitude-based pruning operates solely on weight values, ignoring critical nonlinear interactions within the model. This limitation is particularly impactful in prompt-based tuning, where the prompts are prepended to the input and play a disproportionately large role in the model's behavior compared to standard weights.

### 5.4.3 Comparison with Method Involving Backpropagation

In this section, we compare our method with black-box tuning baselines that involve backpropagation, such as CraFT [43]. The CraFT method introduces a collaborative fine-tuning framework that jointly optimizes both the prompt and the adapter, using ES for the former and first-order (FO) methods for the latter. Although CraFT achieves strong performance, its requirement of backpropagation limits its applicability in memory-constrained environments, such as mobile devices and edge devices, where gradient access via backpropagation is not available.

It is important to highlight that reliance on backpropagation presents significant challenges for deployment on edge inference devices, as these devices are typically equipped with inference-only ASICs and do not support gradient computation. Unlike large-scale multimodal models like LLaVA [25], the CLIP model is particularly relevant for edge computing scenarios. Despite the inherent limitations of backpropagation-based methods in such contexts, we include CraFT in our comparison to demonstrate that our proposed SharpZO method can achieve even better performance without relying on backpropagation, and in a more efficient manner.

The comparison results on few-shot task accuracy and training memory, are summarized in Table 4. As shown, SharpZO consistently outperforms CraFT across all evaluated tasks. Moreover, SharpZO also requires less training memory, as it avoids storing backpropagation graphs, further enhancing its suitability for edge deployment.

Table 4: Comparison between SharpZO and black-box fine-tuning baseline involving backpropagation. * represent accuracy results obtained from original CraFT paper [43]. We bold the best results during the compasion.

| Methods | Test Accuracy | | | | Memory (MB) | | | |
|---------|---------------|------|------|------|-------------|------|------|------|
|         | Imagenet | Pets | DTD | Euro | Imagenet | Pets | DTD | Euro |
| CraFT* | 68.21 | 91.94 | 63.28 | 72.07 | 3297.5 | 3130.9 | 3128.7 | 2178.8 |
| **SharpZO** | **71.60** | **93.46** | **63.95** | **79.42** | **3132.3** | **3032.5** | **3057.9** | **2075.2** |

## 6 Conclusion

This paper has introduced SharpZO, a hybrid ZO fine-tuning method comprising two optimization stages. In Stage 1, SharpZO employs a sharpness-aware CMA-ES algorithm to conduct a global search for optimal regions while simultaneously smoothing the loss landscape. In Stage 2, SharpZO performs fine-grained sparse ZO optimization for local optimization. Compared with prior BP-free fine-tuning approaches, SharpZO provides a high-performance, inference-only fine-tuning solution tailored for VLMs. Future work may explore the extension of SharpZO to full-model fine-tuning for both text- and multimodal LLMs.

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

# A Limitations

While SharpZO demonstrates strong empirical and theoretical advantages for forward-only VLM fine-tuning, several limitations remain. First, the method is currently tailored for prompt-tuning scenarios with relatively low-dimensional parameter spaces; its scalability to full-model or multi-modal fine-tuning remains unexplored. Second, the sharpness-aware CMA-ES warm-up stage requires coordinate-wise gradient estimation (CGE), which may be computationally expensive for higher-dimensional settings. As a result, the SharpZO method proposed in this work is a better fit under the parameter-efficient fine-tuning setup.

# B Further Detail Regarding Tasks and Baselines

**Datasets:** Following the experimental setup of prior VLMs fine-tuning works [43, 32], we evaluate SharpZO on 11 diverse image classification benchmarks under a few-shot learning scenario. These datasets cover a broad range of tasks: generic object recognition with ImageNet [9] and Caltech101 [42], fine-grained image classification with OxfordPets [33], StanfordCars [23], Flowers102 [30], Food101 [4], and FGVCAircraft [27], satellite image classification with EuroSAT [20], texture recognition with DTD [6], scene classification with SUN397 [44], and action recognition with UCF101 [36]. To assess the robustness of SharpZO under distribution shift, we further evaluate it on four widely-used out-of-distribution (OOD) variants of ImageNet: ImageNetV2 [35], ImageNet-Sketch [42], ImageNet-A [22], and ImageNet-R [21].

**Baselines:** To benchmark the performance of SharpZO against SOTA methods, we mianly consider five baseline approaches:

- **Zero-shot (ZS)**: This baseline uses manually crafted prompts to directly evaluate the pretrained CLIP model without any additional adaptation.
- **BBT** [49]: BBT employs a naive CMA-ES-based optimizer to update the trainable prompt parameters. As the original BBT is designed for LLMs, we adopt its prompt generator structure and adapt it to the VLM fine-tuning setting.
- **BlackVIP** [31]: BlackVIP uses a naive ZO-RGE estimator to jointly optimize both textual and visual prompts in a black-box manner.
- **ZIP** [32]: ZIP improves upon naive ZO prompt tuning by reducing the number of trainable parameters via low-rank decomposition of the prompt space.
- **CraFT** [43]: CraFT introduces a trainable adapter appended to the output of the CLIP model. It jointly optimizes both the prompt parameters and the adapter using a combination of CMA-ES and gradient-based methods. As CraFT requires access to backpropagation, we provide a separate comparison with it in Appendix 5.4.3.

# C Additional Experimental Results

## C.1 Robustness across different model architectures

We further evaluate the performance of SharpZO across different model architectures and compare it with other baselines, with results summarized in Table 5. As shown, SharpZO demonstrates architecture-agnostic effectiveness, consistently outperforming previous backpropagation-free (BP-free) methods across all four evaluated architectures. In particular, SharpZO achieves an average absolute performance improvement of **2.25%** and **4.19%** over the ZIP and BlackVIP methods, respectively.

Table 5: Ablation study for model architectures with Imagenet dataset.

| Methods | RN50 | RN101 | Vit-B/16 | Vit-B/32 | Avg. |
|---|---|---|---|---|---|
| ZS-CLIP | 58.18 | 61.62 | 66.73 | 62.05 | 62.15 |
| BlackVIP | 60.33 | 62.00 | 67.10 | 61.10 | 62.63 |
| ZIP | 61.30 | 63.67 | 68.35 | 64.97 | 64.57 |
| **SharpZO** | **63.29** | **65.40** | **71.60** | **66.98** | **66.82** |

## C.2  Hyper-parameter Search

To guide future applications of the SharpZO method, we conduct ablation studies on several key hyper-parameters, including the scaling factor for the sharpness term, and the sparsity ratio used in sparse ZO optimization. The results are summarized in Table 6.

Based on our experiments, the optimal scaling factor for the sharpness term should be around 0.05 or 0.1, which is consistent with the choice in the original SAM paper for the SAM-SGD algorithm. The sparsity ratio should be larger than 0.5. Conversely, a sparsity ratio that is too small will hurt the representational capacity of the prompt parameters, since the number of training parameters is already low enough during prompt tuning. This observation differs from previous ZO full-model fine-tuning work.

Table 6: Ablation studies for different applicable pruning metrics and sparsity.

| Methods\Sparsity | 10% | 30% | 50% | 70% | 90% |
|---|---|---|---|---|---|
| Magnitude | 78.07 | 77.14 | 76.12 | 77.06 | 77.09 |
| Z-Score | 78.91 | 79.02 | 79.42 | 79.15 | 79.21 |

Table 7: Ablation studies for different scaling factor $\rho$.

| $\rho$ | 0.001 | 0.01 | 0.05 | 0.1 | 0.5 |
|---|---|---|---|---|---|
| Accuracy | 77.74 | 79.03 | 79.22 | 79.42 | 76.32 |

Another hyperparameter that influences performance is the scaling factor $\rho$, which adjusts the weight of the sharpness-aware term during the sampling process in Stage 1 optimization. We conduct an ablation study with different values of $\rho$, as shown in Table 7.

We outline the configuration details for each comparative baselines. Specifically, the hyper-parameter setup of individual tasks for SharpZO method are presented in Table 8. To search these hyper-parameters, we select 3-5 candidate values ($[0.1, 0.2, 0.4, 1.0]$ for CMA_ES step size, $[1e-3, 1e-4, 1e-5]$ for ZO scale and $[1e-1, 1e-3, 1e-5]$ for ZO learning rate) and choose the one yielding the best performance with the other hyper-parameters fixed. All hyper-parameter search are performed on a 5-shot validation set extracted from the official validation set or splitted from the training set (e.g. ImageNet). For a fair comparison, we use the **original** hyperparameter settings provided in the baseline papers [32, 31, 43] when running the experiments. In contrast, for the ablation studies, we adopt a consistent parameter setup across methods to ensure comparability. Specifically, we would like to note that during the stage 2 of our method, we set the query number $q$ as 1 instead of 5 used in previous baselines like ZIP and BlackVIP, which is enough for the convergence of SharpZO method.

Table 8: Hyper-parameter setup for Stage 1 and Stage 2 in the SharpZO paper.

| Method | Dataset | Pets | Flo | FGVC | DTD | Euro | Cars | Food | SUN | Cal | UCF | IN |
|---|---|---|---|---|---|---|---|---|---|---|---|---|
| | Step size $\sigma$ | 0.1 | 0.4 | 0.2 | 0.4 | 0.4 | 1.0 | 0.1 | 0.4 | 0.4 | 0.1 | 1.0 |
| | ZO-CGE scale $\mu_{cge}$ | 1e-3 | 1e-3 | 1e-3 | 1e-5 | 1e-3 | 1e-3 | 1e-3 | 1e-3 | 1e-3 | 1e-3 | 1e-5 |
| | Implicit population $S$ | | | | | | 40 | | | | | |
| SharpZO | Intrinsic dimension $d$ | | | | | | 512 | | | | | |
| (Stage 1) | Context tokens number $m$ | | | | | | 4 | | | | | |
| | Scaling factor $\rho$ | | | | | | 0.1 | | | | | |
| | Population size $S$ | | | | | | 40 | | | | | |
| | Change point (step) | 100 | 500 | 400 | 300 | 500 | 100 | 100 | 400 | 200 | 200 | 200 |
| | Learning rate $\eta$ | 1e-3 | 1e-3 | 1e-3 | 1e-3 | 1e-3 | 1e-3 | 1e-3 | 1e-1 | 1e-3 | 1e-3 | 1e-1 |
| | ZO-CGE scale $\mu_{cge}$ | | | | | | 1e-5 | | | | | |
| SharpZO | ZO-RGE scale $\mu_{rge}$ | | | | | | 1e-3 | | | | | |
| (Stage 2) | Pruning interval $K$ | | | | | | 200 | | | | | |
| | Number of query $q$ | | | | | | 1 | | | | | |
| | Pruning ratio | | | | | | 0.5 | | | | | |

# D  Proofs

To prove Theorem 1, we begin by establishing Lemma 1, which characterizes the convergence of the first-stage sharpness-aware CMA-ES method by leveraging its interpretation as a natural-gradient descent algorithm [1]. We then apply the result of Lemma 1 as the initial condition for analyzing the convergence of the second-stage ZO optimization. Finally, by composing these two phases, we obtain the overall convergence rate of the SharpZO algorithm.

## D.1 Proof of Lemma 1

Before presenting the Lemma 1, we first introduce some background knowledge regarding the connection between the CMA-ES update and the natural gradient descent, details about the mathematical relationship can be refereed to [1, 16].

Considering the minimization

$$\arg\min_{\boldsymbol{\theta}_t, \boldsymbol{C}_t} \mathbb{E}\big[\mathcal{L}(P)\,\big|\,\boldsymbol{\theta}_t,\,\boldsymbol{C}_t\big]$$

under the sampling distribution of eq. (3), a natural-gradient descent step with step-size $\sigma$ reads

$$\theta_{t+1} = \boldsymbol{\theta}_t - \sigma\, \mathbf{F}_{\boldsymbol{\theta}_t}^{-1}\, \nabla_{\boldsymbol{\theta}_t}\mathbb{E}\big[\mathcal{L}(P)\,\big|\,\boldsymbol{\theta}_t,\,\boldsymbol{C}_t\big], \tag{7}$$

$$\boldsymbol{C}_{t+1} = \boldsymbol{C}_t - \sigma\, \nabla_{\boldsymbol{C}_t}\mathbb{E}\big[\mathcal{L}(P)\,\big|\,\boldsymbol{\theta}_t,\,\boldsymbol{C}_t\big]. \tag{8}$$

where $F_t^{-1}$ is the Fisher information matrix $\mathbf{F}_{\boldsymbol{\theta}_t} = -\mathbb{E}_{P\sim f(\cdot|\boldsymbol{\theta}_t)}\left[\frac{\partial^2 \log f(P|\boldsymbol{\theta}_t)}{\partial\theta\,\partial\theta^\top}\right]$. given the sample distribution defined in eq. (3). Here we have used that the natural gradient with respect to $\boldsymbol{\theta}_t$ satisfies: $\widetilde{\nabla}_{\boldsymbol{\theta}_t}\mathbb{E}[\mathcal{L}(P)] = \boldsymbol{C}_t^{-1}\nabla_{\boldsymbol{\theta}_t}\mathbb{E}[\mathcal{L}(P)]$.

To simplify the subsequent convergence proof, we note that our target is the expected fitness $f(m_t)$ of the sample center $m_t = \boldsymbol{\theta}_t$, whereas $\boldsymbol{C}_t$ affects only the sampling spread and not directly the objective value. In the idealized infinite-samples regime of CMA-ES one shows

$$\boldsymbol{C}_t^{-1} \ \propto\ \nabla^2\mathcal{L}(m_t) \ =\ H,$$

so that $\boldsymbol{C}_t$ implements a Hessian-inverse preconditioner. Consequently, in our proof we focus solely on the mean update (7) (with $\boldsymbol{C}_t^{-1} \propto H$) and omit carrying the detailed covariance dynamics (8) through the convergence bounds.

Based on eq. (3), we sample $P_i$ based on both the current mean value of the distribution parameter $\boldsymbol{\theta}_t$ and the sharpness aware term $\epsilon^*$ obtained by optimizing the maximize problem $\max_{\|\epsilon\|_2\leq\rho}\mathcal{L}(P+\epsilon)$ within the nearly region around the current parameter $P$. Thus, by simplifying the second and higher order term with the variance of $z$, we can obtain the gradient of expectation for the loss $\mathbb{E}[\mathcal{L}(\boldsymbol{\theta}_t + \epsilon^* + z)]$ as:

$$\nabla_{\boldsymbol{\theta}_t}\mathbb{E}[\mathcal{L}(\boldsymbol{\theta}_t + \epsilon^* + z)] = \nabla_{\boldsymbol{\theta}_t}\mathcal{L}(\boldsymbol{\theta}_t + \epsilon^*) + \mathcal{O}(\delta_t^2)$$

Putting the above equation into eq. (7), we can obtain the natural gradient updating equation for the stage 1 of our SharpZO method, gives:

$$\begin{aligned}
\theta_{t+1} &= \boldsymbol{\theta}_t - \sigma\, \mathbf{F}_{\boldsymbol{\theta}_t}^{-1}\, \nabla_{\boldsymbol{\theta}_t}\mathbb{E}\big[\mathcal{L}(P)\,\big|\,\boldsymbol{\theta}_t,\,\boldsymbol{C}_t\big]\\
&= \boldsymbol{\theta}_t - \sigma\mathbf{F}_{\boldsymbol{\theta}_t}^{-1}[\nabla\mathcal{L}(\boldsymbol{\theta}_t + \epsilon^*) + \mathcal{O}(\delta_t^2)]
\end{aligned}$$

Here, inspired by the proof of Therorem 4.1 of original SAM paper [3], we divide the natural gradient step of our sharpness-aware CMA-ES method into two steps:

$$\theta_{t+\frac{1}{2}} = \boldsymbol{\theta}_t + \rho\frac{\nabla\mathcal{L}(\boldsymbol{\theta}_t)}{\|\nabla\mathcal{L}(\boldsymbol{\theta}_t)\|_2}, \tag{9}$$

$$\theta_{t+1} = \boldsymbol{\theta}_t - \sigma\mathbf{F}_{\boldsymbol{\theta}_t}^{-1}[\nabla\mathcal{L}(\theta_{t+\frac{1}{2}}) + \mathcal{O}(\delta_t^2)] \tag{10}$$

**Lemma 1** (Per–step error bound for sharpness-aware CMA-ES). *Under Assumptions A2, the sharpness-aware CMA-ES method gives a per-step error bound for the updating process as:*

$$\mathcal{L}(\theta_{t+1}) \leq \mathcal{L}(\boldsymbol{\theta}_t) - \tfrac{1}{2}(1 - 2L\sigma^2)\|\nabla'\mathcal{L}(\boldsymbol{\theta}_t)\|^2 + L^2\rho^2 \ + \ L^3\sigma^2\,\rho^2$$

*Proof.* We begin our proof for the first stage defined in eq. (9). By the assumption of L-smoothness, we have:

$$\mathcal{L}(\theta_{t+\frac{1}{2}}) \leq \mathcal{L}(\boldsymbol{\theta}_t) + \big\langle\nabla\mathcal{L}(\boldsymbol{\theta}_t),\, \theta_{t+\frac{1}{2}} - \boldsymbol{\theta}_t\big\rangle + \tfrac{L}{2}\|\theta_{t+\frac{1}{2}} - \boldsymbol{\theta}_t\|^2.$$

Since $\theta_{t+\frac{1}{2}} - \boldsymbol{\theta}_t = \rho\,\nabla\mathcal{L}(\boldsymbol{\theta}_t)/\|\nabla\mathcal{L}(\boldsymbol{\theta}_t)\|$,

$$\mathcal{L}(\theta_{t+\frac{1}{2}}) \leq \mathcal{L}(\boldsymbol{\theta}_t) + \rho\,\|\nabla\mathcal{L}(\boldsymbol{\theta}_t)\| + \tfrac{L\rho^2}{2}. \tag{1}$$

For the second stage in eq. (9), we employ the similar idea and denote natural gradient as $\nabla'\mathcal{L}(\theta_t) \approx \mathbf{F}_{\theta_t}^{-1}[\nabla\mathcal{L}(\theta_t)]$, which gives:

$$
\begin{aligned}
\mathcal{L}(\theta_{t+1}) &\leq \mathcal{L}(\boldsymbol{\theta}_t) - \sigma\left\langle \nabla'\mathcal{L}(\boldsymbol{\theta}_t), \nabla\mathcal{L}(\theta_{t+\frac{1}{2}})\right\rangle + \tfrac{L\sigma^2}{2}\left\|\nabla\mathcal{L}(\theta_{t+\frac{1}{2}})\right\|^2 \\
&\overset{(a)}{\leq} \mathcal{L}(\boldsymbol{\theta}_t) - \tfrac{1}{2}\|\nabla'\mathcal{L}(\boldsymbol{\theta}_t)\|^2 + \tfrac{1}{2}\left\|\nabla'\mathcal{L}(\boldsymbol{\theta}_t) - \nabla\mathcal{L}(\theta_{t+\frac{1}{2}})\right\|^2 + \tfrac{L\sigma^2}{2}\left\|\nabla\mathcal{L}(\theta_{t+\frac{1}{2}})\right\|^2 \\
&\overset{(b)}{\leq} \mathcal{L}(\boldsymbol{\theta}_t) - \tfrac{1}{2}\|\nabla'\mathcal{L}(\boldsymbol{\theta}_t)\|^2 + L^2\rho^2 + \tfrac{L\sigma^2}{2}\left\|\nabla\mathcal{L}(\theta_{t+\frac{1}{2}})\right\|^2 \\
&\leq \mathcal{L}(\boldsymbol{\theta}_t) - \tfrac{1}{2}\|\nabla'\mathcal{L}(\boldsymbol{\theta}_t)\|^2 + L^2\rho^2 + L\sigma^2\|\nabla'\mathcal{L}(\boldsymbol{\theta}_t)\|^2 + L^3\sigma^2\rho^2 \\
&\leq \mathcal{L}(\boldsymbol{\theta}_t) - \tfrac{1}{2}(1 - 2L\sigma^2)\|\nabla'\mathcal{L}(\boldsymbol{\theta}_t)\|^2 + L^2\rho^2 + L^3\sigma^2\rho^2
\end{aligned}
$$

where (a) is given by the fact $\langle a, b\rangle \geq \tfrac{1}{2}\|a\|^2 - \tfrac{1}{2}\|a-b\|^2$ with $a = \nabla\mathcal{L}(\boldsymbol{\theta}_t)$, $b = \nabla\mathcal{L}(\theta_{t+\frac{1}{2}})$ and (b) is given by eq. (9). $\qquad\square$

### D.2 Proof of Theorem 1

Now, we begin the proof of the global convergence rate of SharpZO method. Before we start, we first prove a per-step error bound for the stage 2 sparse ZO training in Lemma 2. Then, we perform inductive step based on the per-step error bound of the stage 2 and include the results in Lemma 1 as an initialization point of stage 2 training. Different from the proof in previous ZO fine-tuning paper [27] that consider an 'effective' rank for the dimension fo the optimization problem, we consider the true dimension $d$, as the trainable parameter in our prompt tuning case is much lower than the full model fine-tuning case. The proof of Lemma 2 is given as follows:

**Lemma 2** (Per-step Error Bound for ZO-SGD). *Given the Assumption A2 and the ZO-RGE gradient estimation follow eq. (4), by setting the learning rate $\eta \leq \frac{1}{2L(d+4)}$, we have:*

$$
\mathbb{E}[\mathcal{L}(\boldsymbol{w}_{t+1}) \mid \boldsymbol{w}_t] \leq \mathcal{L}(\boldsymbol{w}_t) - \frac{\eta}{2}\|\nabla\mathcal{L}(\boldsymbol{w}_t)\|^2 + L\eta^2\left(\gamma^2(d+4) + \frac{L^2\mu^2}{4}(d+6)^3\right),
$$

*where $d$ is the true parameter dimension of the trainable prompt $\boldsymbol{w}$ and $L$ is the smoothness factor, $\mu$ is the ZO perturbation scal. The standard devation of the stochastic gradient estimation $\gamma_t$ is defined as $\gamma_t = \mathbb{E}[\|\hat{\nabla}\mathcal{L}(\boldsymbol{w}_t) - \mathcal{L}_\mu(\boldsymbol{w}_t)\|^2]$, given the unbiased estimator $\hat{\nabla}\mathcal{L}(\boldsymbol{w}_t)$ for the smoothed objective function $\mathcal{L}_\mu(\boldsymbol{w})$.*

*Proof.* Let $\boldsymbol{w}_t$ be the parameter at iteration $t$ and we consider ZO-SGD using a gaussian smoothing estimator defined in eq. (4). Based on properties of $\mathcal{L}_\mu$ Theorem 3.1 (c) of [], the variance of the estimator satisfies:

$$
\mathbb{E}[\|\hat{\nabla}\mathcal{L}(\boldsymbol{w}_t)\|^2] \leq 2(d+4)[\|\nabla\mathcal{L}(\boldsymbol{w}_t)\|^2 + \gamma^2] + \frac{\mu^2}{2}L^2(n+6)^3,
$$

where is smoothness factor $L$ is assumed

Given the learning rate $\eta > 0$, then from the smoothness of $\mathcal{L}$, the standard descent lemma gives:

$$
\begin{aligned}
\mathbb{E}[\mathcal{L}(\boldsymbol{w}_{t+1}) \mid \boldsymbol{w}_t] &\leq \mathcal{L}(\boldsymbol{w}_t) - \eta\|\nabla\mathcal{L}(\boldsymbol{w}_t)\|^2 + \frac{L}{2}\eta^2\mathbb{E}[\|\hat{\nabla}\mathcal{L}(\boldsymbol{w}_t)\|^2] \\
&\leq \mathcal{L}(\boldsymbol{w}_t) - \eta\|\nabla\mathcal{L}(\boldsymbol{w}_t)\|^2 + \frac{L}{2}\eta^2\left(2(d+4)[\|\nabla\mathcal{L}(\boldsymbol{w}_t)\|^2 + \gamma^2] + \frac{\mu^2}{2}L^2(d+6)^3\right)
\end{aligned}
$$

Rearranging:

$$
\mathbb{E}[\mathcal{L}(\boldsymbol{w}_{t+1}) \mid \boldsymbol{w}_t] \leq \mathcal{L}(\boldsymbol{w}_t) - \eta\left(1 - L\eta(d+4)\right)\|\nabla\mathcal{L}(\boldsymbol{w}_t)\|^2 + L\eta^2\left(\gamma^2(d+4) + \frac{L^2\mu^2}{4}(d+6)^3\right).
$$

Choose $\eta \leq \frac{1}{2L(d+4)}$ so that $1 - L\eta(d+4) \geq \frac{1}{2}$. Then:

$$
\mathbb{E}[\mathcal{L}(\boldsymbol{w}_{t+1}) \mid \boldsymbol{w}_t] \leq \mathcal{L}(\boldsymbol{w}_t) - \frac{\eta}{2}\|\nabla\mathcal{L}(\boldsymbol{w}_t)\|^2 + L\eta^2\left(\gamma^2(d+4) + \frac{L^2\mu^2}{4}(d+6)^3\right).
$$

$\qquad\square$

Next, we proceed to prove Theorem 1 by performing an inductive argument based on the result of Lemma 2. Let the total number of steps in Stage 2 be denoted as $T_2 := T - T_c$. To facilitate the analysis, we define two suboptimality gap measures:

- $\Delta_t^{(1)} := \mathcal{L}(\boldsymbol{\theta}_t) - \mathcal{L}^*$, which denotes the optimality gap of the *distributional mean* $\boldsymbol{\theta}_t$ used in Stage 1 (Sharpness-aware CMA-ES);
- $\Delta_t^{(2)} := \mathcal{L}(\boldsymbol{w}_t) - \mathcal{L}^*$ used in Stage 2 (ZO optimization).

At the transition point $t = T_c$, Lemma 1 guarantees that the distributional mean $\theta_{T_c}$ satisfies a convergence bound on $\Delta_{T_c}^{(1)}$. Using a second-order Taylor expansion of the loss function around $\theta_{T_c}$, we can relate the Stage 2 initialization gap $\Delta_{T_c}^{(2)}$ to $\Delta_{T_c}^{(1)}$ via:

$$\Delta_{T_c}^{(2)} = \mathcal{L}(p_{T_c}) - \mathcal{L}^* \leq \Delta_{T_c}^{(1)} + \rho \|\nabla \mathcal{L}(\theta_{T_c})\| + \frac{L}{2}\left(\rho^2 + \delta_{T_c}^2 \operatorname{Tr}(C_{T_c})\right). \tag{11}$$

This inequality provides the initial condition for the inductive proof in Stage 2, where we now track the evolution of $\Delta_t^{(2)}$ for $t = T_c, \ldots, T$, as governed by Lemma 2. Here, we first bound $\Delta_{T_c}^{(1)}$

By Lemma 1, for each $t$ we have

$$\mathcal{L}(\boldsymbol{\theta}_{t+1}) \ \leq \ \mathcal{L}(\boldsymbol{\theta}_t) \ - \ \tfrac{1}{2}\left(1 - 2L\sigma^2\right)\|\nabla \mathcal{L}(\boldsymbol{\theta}_t)\|^2 + L^2\rho^2 + L^3\sigma^2\rho^2.$$

Subtracting $\mathcal{L}^*$ from both sides yields

$$\Delta_{t+1} \ \leq \ \Delta_t \ - \ \tfrac{1}{2}\left(1 - 2L\sigma^2\right)\|\nabla \mathcal{L}(\boldsymbol{\theta}_t)\|^2 + C, \quad C := L^2\rho^2 + L^3\sigma^2\rho^2.$$

Under the PL inequality $\|\nabla \mathcal{L}(\boldsymbol{\theta}_t)\|^2 \geq 2\mu\,\Delta_t$, it follows that

$$\begin{aligned}\Delta_{t+1}^{(1)} &\leq \Delta_t^{(1)} - \tfrac{1}{2}\left(1 - 2L\sigma^2\right)(2\mu\,\Delta_t) + C \\ &= \left[1 - \mu(1 - 2L\sigma^2)\right]\Delta_t + C.\end{aligned}$$

Set

$$\xi \ := \ \mu\left(1 - 2L\sigma^2\right), \quad 0 < \xi < 1 \quad (\text{assuming } \sigma^2 < 1/(2L)).$$

Then the recursion becomes

$$\Delta_{t+1}^{(1)} \ \leq \ (1 - \xi)\,\Delta_t^{(1)} + C.$$

Unrolling this for $t = 0, 1, \ldots, T_c - 1$ gives

$$\Delta_{T_c}^{(1)} \ \leq \ (1 - \xi)^{T_c}\Delta_0^{(1)} + C \sum_{i=0}^{T_c-1}(1 - \xi)^i \ = \ (1 - \xi)^{T_c}\Delta_0^{(1)} + \frac{C}{\xi}\left[1 - (1 - \xi)^{T_c}\right].$$

Substituting back $C = L^2\rho^2 + L^3\sigma^2\rho^2$ and $\xi = \mu(1 - 2L\sigma^2)$ completes the proof:

$$\Delta_{T_c}^{(1)} \ \leq \ (1 - \xi)^{T_c}\Delta_0^{(1)} \ + \ \frac{L^2\rho^2 + L^3\sigma^2\rho^2}{\mu\left(1 - 2L\sigma^2\right)}\left[1 - (1 - \xi)^{T_c}\right]. \tag{12}$$

Now, we begin to prove the global convergence rate for the SharpZO method. Let's focus back into the bound given in Lemma 2. By the PL-inequality assumed in Assumption A1, we have:

$$\begin{aligned}\mathbb{E}\left[\mathcal{L}(\boldsymbol{w}_{t+1}) \mid \boldsymbol{w}_t\right] &\leq \mathcal{L}(\boldsymbol{w}_t) - \frac{\eta}{2}\|\nabla \mathcal{L}(\boldsymbol{w}_t)\|^2 + C_{\text{var}}(d, L)\,\eta^2 \\ &\leq \mathcal{L}(\boldsymbol{w}_t) - \eta\mu\left(\mathcal{L}(\boldsymbol{w}_t) - \mathcal{L}^*\right) + C_{\text{var}}(d, L)\,\eta^2,\end{aligned}$$

where $C_{\text{var}}(d, L) = \frac{L(d+4) + \frac{L^2\mu^2}{4}(d+6)^3}{\eta}$ is some constant. Taking full expectation and with $\Delta_t^{(2)} := \mathbb{E}[\mathcal{L}(\boldsymbol{w}_t)] - \mathcal{L}^*$ gives the one-step contraction

$$\Delta_{t+1}^{(2)} \ \leq \ (1 - \eta\mu)\,\Delta_t^{(2)} \ + \ C_{\text{var}}(d, L)\,\eta^2.$$

Given the current step $t$, we define $t_2 := t - T_c$ and unroll this linear recursion for $t = T_c, \ldots, T_c + t_2$ yields

$$\Delta_t^{(2)} \leq (1 - \eta\mu)^{t_2} \, \Delta_{T_c}^{(2)} + \frac{C_{\mathrm{var}}(d, L)\,\eta}{\mu} \left( 1 - (1 - \eta\mu)^{t_2} \right)$$

$$\overset{(a)}{\leq} (1 - \eta\mu)^{t_2} (\Delta_{T_c}^{(1)} + \rho\|\nabla\mathcal{L}(\theta_{T_c})\| + \frac{L}{2} \left( \rho^2 + \delta_{T_c}^2 \, \mathrm{Tr}(C_{T_c}) \right)) + \frac{C_{\mathrm{var}}(d, L)\,\eta}{\mu} \left( 1 - (1 - \eta\mu)^{t_2} \right)$$

$$\overset{(b)}{\leq} (1 - \eta\mu)^{t_2} ((1 - \xi)^{T_c} \, \Delta_0^{(1)} \ + \ \frac{L^2\rho^2 + L^3\eta^2\rho^2}{\mu(1 - 2L\eta^2)} \left[ 1 - (1 - \xi)^{T_c} \right]$$

$$+ \rho\|\nabla\mathcal{L}(\theta_{T_c})\| + \frac{L}{2} \left( \rho^2 + \delta_{T_c}^2 \, \mathrm{Tr}(C_{T_c}) \right)) + \frac{C_{\mathrm{var}}(d, L)\,\eta}{\mu} \left( 1 - (1 - \eta\mu)^{t_2} \right),$$

where (a) is given by eq. (11) and (b) follows the bound of $\Delta_{T_c}^{(1)}$ proved in eq. (12). Finally, by ensuring $\Delta_t^{(2)} \leq \epsilon$, we have:

$$t_2 = \mathcal{O}(\frac{1}{\eta\mu} \ln \frac{X - B}{\epsilon - B}),$$

where

$$X = (1 - \xi)^{T_c} \, \Delta_0^{(1)} + \frac{L^2\rho^2 + L^3\eta^2\rho^2}{\xi} \left[ 1 - (1 - \xi)^{T_c} \right] + \rho\|\nabla\mathcal{L}(\theta_{T_c})\| + \frac{L}{2}(\rho^2 + \delta_{T_c}^2 \, \mathrm{Tr}\, C_{T_c})$$

$$\tag{13}$$

$$= (1 - \xi)^{T_c} \, \Delta_0^{(1)} + \rho^2((L^2 + L^3\eta^2)(\xi - 2) + \frac{1}{2}L) + \rho\|\nabla\mathcal{L}(\theta_{T_c})\| + \frac{L\delta_{T_c}^2 \, \mathrm{Tr}\, C_{T_c}}{2} \tag{14}$$

and

$$B = \frac{C_{\mathrm{var}}(d, L)\,\eta}{\mu}, \quad C_{\mathrm{var}}(d, L) = \frac{L(d+4) + \frac{L^2\mu^2}{4}(d+6)^3}{\eta} \tag{15}$$

Here, if we focus on the influence of smoothness factor $L$ and ignoring the lower-order terms for convenience, we can write the convergence rate $t_2$ as:

$$t_2(L) \approx \mathcal{O}\left( \frac{1}{\eta\mu} \log \left( \frac{L^3\eta^2\rho^2}{\epsilon} \right) \right) \tag{16}$$

