# OpenReview forum: "SharpZO: Hybrid Sharpness-Aware Vision Language Model Prompt Tuning via Forward-Only Passes"
_NeurIPS.cc/2025/Conference — NeurIPS 2025 poster_

### Official Review · Reviewer_ZKiu · 2025-06-10

**Clarity:** 3
**Significance:** 3
**Originality:** 3
**Rating:** 4
**Confidence:** 3

**Summary:**

This paper proposes SharpZO, a hybrid sharpness-aware zeroth-order (ZO) optimization method designed for fine-tuning CLIP, using only forward passes. The work is motivated by the limitations of BP-free tuning in resource-constrained environments.  This approach is entirely forward-pass based, making it attractive for inference-only settings. The authors support their method with theoretical convergence analysis and extensive empirical validation across 11 datasets, showing strong performance improvements over previous BP-free methods.

**Questions:**

1. How were the training and test datasets split? Did the authors follow the same protocol as CoOp?

**Ethical Concerns:**

["NO or VERY MINOR ethics concerns only"]

**Final Justification:**

Although I am concerned about the use of a validation set in the few-shot setting, which I believe contradicts the premise of limited labeled data in this setting, the technical contributions of this paper seems sufficient to me. Therefore, I maintain my positive score of 4.

**Limitations:**

Yes

**Quality:**

3

**Strengths And Weaknesses:**

Strengths:

1. The paper provides a solid theoretical analysis that offers valuable insights into the convergence behavior of SharpZO.

2. The experimental evaluation is thorough and well-executed, featuring comparisons with strong baselines, detailed ablation studies, robustness tests under distribution shifts, and efficiency analysis.

Weaknesses:

1. The transition point between training stages is determined using validation accuracy. However, the paper does not clarify how validation accuracy is computed—especially for datasets that lack an official validation split, such as ImageNet. Furthermore, evaluating validation accuracy during training could introduce significant overhead, potentially making the method less suitable for the low-resource settings that this paper aims to support.

2. Although the method is motivated as a general solution for VLMs, the experiments are conducted solely on CLIP. This raises concerns about the generalizability of the proposed approach. The authors are encouraged to evaluate SharpZO on other vision-language models such as LLaVA or SigLIP.

3. The paper focuses primarily on few-shot learning. However, in real-world edge device scenarios, models are often expected to handle distribution shifts across disjoint class sets. A base-to-new generalization setting would better reflect such conditions. Including results in this setting would enhance the practical relevance of the proposed method.

---

> ### Author Rebuttal · Authors · 2025-07-31
>
> Thank you for your thoughtful and detailed feedback. We greatly appreciate your recognition of the strengths of our work, as well as your constructive comments regarding its weaknesses. Below, we address each of the points raised in your review.
>
> ### Weaknesses
>
> > The transition point between training stages is determined using validation accuracy. However, the paper does not clarify how validation accuracy is computed—especially for datasets that lack an official validation split, such as ImageNet. Furthermore, evaluating validation accuracy during training could introduce significant overhead, potentially making the method less suitable for the low-resource settings that this paper aims to support.
> >
>
> For datasets like ImageNet, we manually select validation samples from the training set. In our paper, we adopt a 5-shot per class sampling strategy to construct the validation set. We will add this detail in the revised version.
>
> To ensure that validation evaluation does not become a bottleneck, we use a simple trick: since the prompt weights only affect the text encoder in the CLIP model, we pre-compute and cache all image features for the validation set. This avoids repeated computation of the vision backbone, which is typically the most time-consuming part of inference.
>
> With this trick, a single validation accuracy evaluation only takes **0.009 seconds** taking EuroSAT dataset as an example. Even with 50 evaluations throughout training, the overhead is minimal and does not significantly affect overall efficiency.
>
> > Although the method is motivated as a general solution for VLMs, the experiments are conducted solely on CLIP. This raises concerns about the generalizability of the proposed approach. The authors are encouraged to evaluate SharpZO on other vision-language models such as LLaVA or SigLIP.
> >
>
> Thank you for the suggestion. We are happy to provide additional experimental results using the SigLIP model to demonstrate the generalizability of our method. The experiments follow the same setup as in the ZIP paper to ensure a fair comparison within the limited timeframe of the rebuttal. As observed, the SharpZO method achieves better performance than the ZIP method on the SigLIP model, highlighting its strong generalization ability.
>
> |  | EuroSAT | DTD | OxfordPets |
> | --- | --- | --- | --- |
> | ZIP* | 48.7 | 49.4 | 82.0 |
> | SharpZO | 53.4 | 56.5 | 84.6 |
>
> *result obtained from ZIP paper
>
> > The paper focuses primarily on few-shot learning. However, in real-world edge device scenarios, models are often expected to handle distribution shifts across disjoint class sets. A base-to-new generalization setting would better reflect such conditions. Including results in this setting would enhance the practical relevance of the proposed method.
> >
>
> This is a great point. Following the implementation of the Table 3 of ZIP paper, we train the model on base class and evaluate them on the new classes.  We show some key results here and will expand the comparison in the revised paper:
>
> |  | ImageNet | EuroSAT | DTD | OxfordPets |
> | --- | --- | --- | --- | --- |
> | ZIP* |  65.6 | 64.4 |  51.0 | 97.0 |
> | SharpZO | 67.1 | 70.4 | 53.5 | 95.2 |
>
> *result obtained from ZIP paper
>
> As we can observe, in the base-to-new generalization setting, our SharpZO method still demonstrates a clear advantage compared to SOTA ZO fine-tuning methods in most of the tasks.
>
> ### Questions
>
> > How were the training and test datasets split? Did the authors follow the same protocol as CoOp?
> >
>
> Yes, we follow the same protocol as CoOp for loading dataset.

---

> > ### Comment · Reviewer_ZKiu · 2025-08-01
> > **Thank you for detailed rebuttal**
> >
> > I thank authors for their detailed rebuttal. The rebuttal addresses most of my concerns.
> >
> > I have a new question about the utilization about validation set. Few-shot prompt tuning aims to address labeled data are limited. Therefore, I am not sure whether using a validation is proper. For many datasets, the  size of validation dataset  is even much larger than few-shot training set, which seems not reasonable. Moreover, if I have a validation set, why not use it for training as labeled data is very limited.  Hope to see insightful discussion from authors.

---

> ### Author Response · Authors · 2025-08-01
> **Thanks for your follow-up**
>
> Thank you for the follow-up from the reviewer. Actually, we had considered this issue earlier, which is why we mentioned that we adopt a 5-shot per class sampling strategy to construct the validation set in previous rebuttal. Typically, we sample from the validation dataset, except for ImageNet, where we sample from the training dataset due to the absence of an official validation split.
>
> Additionally, we would like to note that the number of shots in the validation set should not significantly affect the results of the SharpZO method. Since selecting the change point is more relaxed compared to choosing the best checkpoint, we do not need to pick a “perfect” change point. The stage 2 optimization can effectively make up the difference if the change point falls within a reasonable range. This observation is also supported by our previous tests with fixed change points.
>
> | EuroSAT ACC | 50 | 100 | Auto-tran at 145 | 150 | 200 |
> | --- | --- | --- | --- | --- | --- |
> | Change Point ACC | 73.26 | 75.24 | 76.58 | 76.34 | 77.87 |
> | Final ACC | 78.20 | 79.65 | 79.42 | 79.46 | 80.27 |
>
> Best,
>
> Authors

---

> > ### Comment · Reviewer_ZKiu · 2025-08-03
> > **Thanks for further information**
> >
> > Thank authors for further clarification. I am actually questioned about the setting of using validation set but not the number. Given a large scale training dataset, using validation set is reasonable to avoid overfitting. But with few-shot setting, using validation is a little strange to me. For example, using them as validation set means they are also involved in training. So why not just use them for training? This I believe will lead to much better results. Of course, I do not mean further experiment results but to share my concerns about the setting.
> >
> > Anyway, I will maintain my positive score and believe the contribution of this paper is enough.

---

> ### Author Response · Authors · 2025-08-03
> **Thank you for the positive evaluation and further clarification**
>
> Thank you for maintaining your positive evaluation and for your further clarification about your question.
>
> Although this is a minor issue, we are still happy to provide more details. As in previous work (e.g., CoOp, as mentioned by reviewer qycS), a validation set is commonly used for hyper-parameter search. Using a validation dataset to determine the change point can be viewed as a form of such hyper-parameter search. There is always a trade-off when using a validation dataset: it will reduce the size of training dataset, but allows us to choose better hyper-parameters.  We used the similar experimental setup of prior work for fair comparison.
>
> To provide more flexibility, we will also offer the option of using a fixed transition point obtained through hyper-parameter search, in case some readers prefer to use the whole dataset for training.
>
>
> Best,
>
> Authors

---

### Official Review · Reviewer_qycS · 2025-06-27

**Clarity:** 4
**Significance:** 3
**Originality:** 3
**Rating:** 5
**Confidence:** 3

**Summary:**

This paper proposes an effective and efficient backpropagation-free (BP-free) prompt tuning method named SharpZO. The method consists of two stages. The first stage employs a sharpness-aware evolutionary strategy that combines coordinate-wise gradient estimation with sharpness-aware optimization to construct a smoothed initialization. The second stage performs fine-grained local optimization via sparse zeroth-order (ZO) optimization using randomized gradient estimation. The authors also provide a theoretical convergence analysis to justify the superiority of the proposed method. Extensive experiments demonstrate that SharpZO significantly outperforms existing BP-free methods in both accuracy and training efficiency, as measured by wall-clock time.

**Questions:**

Please refer to Weaknesses.

**Ethical Concerns:**

["NO or VERY MINOR ethics concerns only"]

**Final Justification:**

All my concerns have been addressed

**Limitations:**

yes

**Paper Formatting Concerns:**

N.A.

**Quality:**

4

**Strengths And Weaknesses:**

Strengths:

1. The paper is well-written and easy to follow. As someone not familiar with BP-free tuning methods, I appreciate that the paper is self-contained and provides sufficient background to understand the key concepts.

2. The proposed method is elegant and intuitively convincing, which makes me believe it is likely to work well in practice. In addition, the authors provide a theoretical convergence analysis to further justify its effectiveness.

3. The empirical improvements are significant in both accuracy and training time, which makes me believe this work has the potential to be influential.

Weaknesses:
I do not find any major weaknesses in the paper. Rather, I have a few questions or suggestions that I would like the authors to response:

1. As someone not from the field of BP-free optimization, I would appreciate a brief introduction to the Covariance Matrix Adaptation Evolution Strategy (CMA-ES) to provide necessary background. (e.g. why we cannot directly optimize the prompt embedding \bar{p}?).

2. In line 143, $C$ should be typeset in boldface.

3. The proposed method is not inherently limited to prompt tuning and could be applied to other tuning frameworks such as linear probing. I would be interested to see how SharpZO performs when combined with linear probing.

4. While SharpZO is a BP-free method and understandably compared with other BP-free baselines, its strong performance warrants additional context (I noticed that its performance is significantly improved). I suggest including results from BP-based methods such as CoOp [1] and ProGrad [2] as reference points to better contextualize the remaining performance gap with BP-based methods. And whether add objective of [2] can further improve the performance of SharpZO?

[1] Learning to Prompt for Vision-Language Models, IJCV 2022.

[2] Prompt-aligned gradient for prompt tuning, ICCV 2023.

To summarize, this is a solid paper with many valuable insights. I am open to raising my score if the authors adequately address the questions and suggestions above.

---

> ### Author Rebuttal · Authors · 2025-07-30
>
> Thank you for your thoughtful and detailed feedback. We greatly appreciate your recognition of the strengths of our work, as well as your constructive comments regarding its weaknesses. Below, we address each of the points raised in your review.
>
> ### Weaknesses
>
> > As someone not from the field of BP-free optimization, I would appreciate a brief introduction to the Covariance Matrix Adaptation Evolution Strategy (CMA-ES) to provide necessary background. (e.g. why we cannot directly optimize the prompt embedding \bar{p}?).
> >
>
> Thank you for the suggestions. In one sentence, CMA-ES treats the trainable parameters as a multivariate normal distribution and updates its parameters (mean and covariance) to search for global optima. We will add a brief introduction to CMA-ES in the Background section of the revised paper.
>
> We update the prompt embeddings in a low-dimensional space to reduce the variance in both Stage 1 and Stage 2 training, as both CMA-ES and ZO optimization exhibit dimension-dependent convergence rates. The effectiveness of reducing variance via low-rank projection has been demonstrated in prior work [1][2].
>
> [1] ZIP: An efficient zeroth-order prompt tuning for black-box vision-language models
>
> [2] Adazeta: Adaptive zeroth-order tensor-train adaption for memory-efficient large language models fine-tuning
>
> > In line 143, should be typeset in boldface.
> >
>
> Thanks for reporting this. We will correct in the final version.
>
> > The proposed method is not inherently limited to prompt tuning and could be applied to other tuning frameworks such as linear probing. I would be interested to see how SharpZO performs when combined with linear probing.
> >
>
> This is an interesting point. To explore this, we attach a classifier on top of the CLIP model to do linear probing with SharpZO (SharpZO-LP) with ZO-SGD (ZO-SGD-LP) and a first-order method using Adam (Adam-LP). The results are summarized as follows:
>
> | Vit-B/16 | EuroSAT | DTD | OxfordPets |
> | --- | --- | --- | --- |
> | Adam-LP | 57.51 | 59.69 | 87.98 |
> | ZO-SGD-LP | 25.67 | 31.98 | 66.51 |
> | SharpZO-LP | 35.55 | 39.77 | 84.98 |
> | SharpZO |  79.42 | 63.95 | 94.06 |
>
> As we can observe, the linear probing is more sensitive to the optimization noise with the shallow and unregularized classifier, compared with prompt tuning. Even though the SharpZO can improve the performance compared with ZO-SGD, there is still a large gap between Adam method.
>
> > While SharpZO is a BP-free method and understandably compared with other BP-free baselines, its strong performance warrants additional context (I noticed that its performance is significantly improved). I suggest including results from BP-based methods such as CoOp [1] and ProGrad [2] as reference points to better contextualize the remaining performance gap with BP-based methods. And whether add objective of [2] can further improve the performance of SharpZO?
> >
>
> Thanks for the suggestion. We agree with your point and will include the BP-based method results in the revised version. We summarize the key comparison results here and will expand the comparison in Table 1 in the revised version:
>
> | RN50 | EuroSAT | DTD | OxfordPets | ImageNet | AVG |
> | --- | --- | --- | --- | --- | --- |
> | CooP | 84.05 | 63.26 | 87.06  | 63.00  | 73.48  |
> | ProGrad | 83.74  | 63.97  | 89.00  | 63.45  | 74.28  |
> | SharpZO | 80.77 | 60.58 | 89.51 | 63.29 | 69.76  |
>
> We can observe the our BP-free SharpZO performs on-par or better than the BP-based prompt tuning method CoOP on dataset like OxfordPets and ImageNet. We also agree there is still a gap between SharpZO and CoOp on other datasets. This is limited by the stochastic nature of zero-order fine-tuning, which in return, bring in the advantage of very low memory cost and inference-only fine-tuning availability.
>
> However, Regarding the combination of the objective proposed in [2] with SharpZO, we believe it may not be appropriate. In SharpZO, we employ ZO-RGE to estimate gradients by assuming a fixed update direction randomly generated at each step and adjusting the magnitude to reflect the quality of that direction. In contrast, the objective in [2] requires computing the angle between two gradients. Since our approach involves randomly generated directions, calculating such angles does not provide meaningful information. Therefore, we believe the training objective in [2] is not well-aligned with the principles of our method and will not provide additional benefit.

---

> ### Comment · Reviewer_qycS · 2025-08-03
>
> Thanks to the authors for their responses, which have addressed all my previous concerns.
>
> However, after reviewing ZKiu’s comment, I agree with the concern regarding the validation set size in few-shot settings. In BP-based CLIP few-shot setups (e.g., 1 to 16 shots), the validation set typically contains no more than 4 shots. Could the authors clarify how many validation shots were used in their experiments?
>
> I checked the code provided in the supplementary material and noticed that it appears to be based on the CoOp codebase. However, it seems that the line controlling the validation set size was removed (or perhaps I missed it). For example:
>
> ```
> val = self.generate_fewshot_dataset(val, num_shots=min(num_shots, 4))
> ```
> is in CoOp's codebase, but I cannot find it in your file.
>
> If the authors used a larger validation set, I would appreciate seeing the results under a 4-shot validation setting for a fair comparison.
>
> ---Edited---
>
> I initially considered requesting this result ("If the authors used a larger validation set, I would appreciate seeing the results under a 4-shot validation setting for a fair comparison. "), but later decided it's sufficient for the authors to clarify the validation setting in the final version.
>
> That said, using a larger validation set does not affect my overall recommendation regarding the acceptance of this paper.  I also appreciate that the authors have promised in the rebuttal to include the relevant content in the revised submission. Since my initial concerns have been addressed, I will raise my score.

---

> > ### Author Response · Authors · 2025-08-03
> > **Thank you for your response**
> >
> > Thank you for acknowledging that our responses have addressed your previous concerns. Regarding your minor concern regarding the validation dataset, we pre-selected a 5-shot validation set for each task before running the training code, which is slightly different from the CoOp implementation. To ensure fairness in the experiments, we also conducted a brief comparison by using 4-shot versus 5-shot validation (with the ViT-B/16 backbone):
> >
> > | EuroSAT | Change Point | Test Accuracy | Caltech101 | Change Point | Test Accuracy |
> > | --- | --- | --- | --- | --- | --- |
> > | 4-shot Val | 146 | 79.47 | 4-shot Val | 112 | 94.86 |
> > | 5-shot Val | 145 | 79.42 | 5-shot Val | 105 | 95.09 |
> >
> > As we can observe, there is not much difference on the change point and final performance by using either 4-shot or 5-shot validation. Additionally, we will also offer the option of using a fixed transition point obtained through hyper-parameter search in the final version, in case some readers prefer to use all data for training.
> >
> > Best,
> >
> > Authors

---

> > > ### Comment · Reviewer_qycS · 2025-08-04
> > >
> > > Thank you for your response. The results appear convincing.

---

### Official Review · Reviewer_yzjx · 2025-07-03

**Clarity:** 3
**Significance:** 4
**Originality:** 4
**Rating:** 5
**Confidence:** 3

**Summary:**

This paper proposes a novel backpropagation-free fine-tuning method for VLM's prompt optimization. Their method is based on a 2-stage hybrid optimization process between an evolutionary strategy, CMA-ES and zero-order (ZO) optimization. They introduce interesting and effectiveness changes to both stages that result in a better performance over the SOTA. Extensive experiments and empirical validations provide strong evidence for the effectiveness of their method.

**Questions:**

1. When estimating $\epsilon*$, using CGE, do you use $w_i$ for the estimation? Which would mean that $\epsilon*$ actually also depends on $i$.
2. While SharpZO only requires 2 queries per step in Stage 2, it would seem like Stage 1 requires a significant number of forward passes and if stage 1 were not fixed to 500 steps, how would the ToTA change? especially in comparison with the other methods.

Minor:
- line 154, is $E[z]$ meant to be $E[o]$?

**Ethical Concerns:**

["NO or VERY MINOR ethics concerns only"]

**Final Justification:**

My initial assessment was already quite positive and the authors have addressed my questions and concerns thoroughly during the rebuttal. I think this is a well-written paper that will contribute greatly to the community and should be accepted.

**Limitations:**

yes

**Quality:**

3

**Strengths And Weaknesses:**

Strengths:
1. This paper is well-written (despite a few minor typos) that has a clear and easy to follow structure. Each part of their method, theory and experiments were explained comprehensively with sufficient background, making it smooth for the reader to follow.
2. Besides the novelty in combining 2 different optimization strategies, ES and ZO, the authors also propose new and effective changes to each stage, further strengthening their method. (Though, I'm not as familiar with other ZO methods, so I defer the final judgement on its novelty to other reviewers and this simply remains my opinion. )
3. There is some theoretical guarantee for the convergence of their method, providing sufficient theoretical justification for their performance gains.
4. Extensive experimental results and ablations are convincing of the effectiveness of SharpZO.

Weaknesses:
1. In providing the theoretical guarantee in section 4, it would be more clear to state the convergence rate of the baseline ZO-SGD as well, this is for the reader to easily understand and believe that your convergence rate is faster.
2. It would be good to provide the average results in Table 2, since you do reference that on average, you increase by 2/2.8% over the SOTA.
3. In lines 272-274, you seem to be referencing the results in section 5.4.1. On the first read through the paper, it might seem out of place since the reader does not know that you will be comparing naive CMA-ES methods to your sharpness aware version. Moving it to a section below 5.4.1 or referencing that you will compare them in a later section would improve the reader experience.

---

> ### Author Rebuttal · Authors · 2025-07-30
>
> Thank you for your thoughtful and detailed feedback. We greatly appreciate your recognition of the strengths of our work, as well as your constructive comments regarding its weaknesses. Below, we address each of the points raised in your review.
>
> ### Weaknesses
>
> > In providing the theoretical guarantee in section 4, it would be more clear to state the convergence rate of the baseline ZO-SGD as well, this is for the reader to easily understand and believe that your convergence rate is faster.
> >
>
> This is a great point. Unlike the traditional way of presenting convergence rates, we highlight the highest-order term of the smoothness factor $L$ to emphasize its influence on zeroth-order optimization. Compared to the baseline ZO-SGD convergence rate $t(L) \approx \mathcal{O}(\frac{L^3\mu^2}{\epsilon})$, our analysis reveals that the smoothness factor $L$ is further reduced by an additional factor (less than 1) related to the first-stage optimization. Since the theoretical number of iterations required for convergence is proportional to the smoothness of the loss landscape, our method demonstrates a faster convergence rate compared to baseline ZO-SGD. We will further discuss these points in the revised version
>
> > It would be good to provide the average results in Table 2, since you do reference that on average, you increase by 2/2.8% over the SOTA.
> >
>
> Thanks for the suggestion. We will include the averaged results to better reflect the overall improvement compared to previous methods.
>
> > In lines 272-274, you seem to be referencing the results in section 5.4.1. On the first read through the paper, it might seem out of place since the reader does not know that you will be comparing naive CMA-ES methods to your sharpness aware version. Moving it to a section below 5.4.1 or referencing that you will compare them in a later section would improve the reader experience.
> >
>
> Thanks for the suggestion. We will move this part of the discussion to Section 5.4.1 to improve the reader's experience.
>
> ### Questions
>
> > When estimating **\epsilon^**using CGE, do you use **w_i** for the estimation? Which would mean that **\epsilon^**actually also depends on **i**.
> >
>
> Sorry for the confusion here. The computation of the perturbation $\epsilon$ should be completed before the CMA-ES sampling at each step, which means the value of trainable matrix $w_t$ should be the same as the latest mean value $\theta_t$. Thus, a better way to present the right part in eq. 3 should be $\epsilon^*=\rho\frac{\nabla\mathcal{L}(w_t)}{\|\nabla\mathcal{L}(w_t)\|_2}$, where $w_t=\theta_t$. So, this computation process don’t depend on $i$. We will make this point clear in the revised version.
>
> > While SharpZO only requires 2 queries per step in Stage 2, it would seem like Stage 1 requires a significant number of forward passes, and if Stage 1 were not fixed to 500 steps, how would the ToTA change? Especially in comparison with the other methods.
> >
>
> We agree stage 1 requires a significant number of forward passes. If we increase the number of Stage 1 step, the ToTA will inevitably exceed that of other methods in some point.
>
> However, this point ( ~1800 steps) is far beyond 500 steps based on our experiments. Here, we provide an example on EuroSAT task. In our setup, the total number of training steps is fixed at 20,000 for all methods. For SharpZO, Stage 1 training takes approximately 5 seconds per step, while Stage 2 training is significantly faster at 0.15 seconds per step. In comparison, ZIP and BlackVIP require roughly 5 times more queries than SharpZO’s Stage 2, resulting in about 0.6 seconds per step, based on our empirical measurements. To ensure SharpZO run slower than ZIP, the switch from Stage 1 to Stage 2 must occur **later than around 1800 steps**.
>
> Also, we would like to clarify that across all 11 datasets we tested, Stage 1 consistently converges within 500 steps. The actual number of change point for some of the tasks determined by the system are listed as below:
>
> |  | ImageNet | caltech101 | eurosat | food101 | stanford_cars |
> | --- | --- | --- | --- | --- | --- |
> | Change Point | 125 | 105 | 145 | 80 | 65 |
>
> ### **Minor:**
>
> > Line 154, is **\mathbb{E}[z]** meant to be **\mathbb{E}[o]**?
> >
>
> Thanks for point this typo out. We will correct it in the revised version.

---

### Official Review · Reviewer_VzLh · 2025-07-03

**Clarity:** 3
**Significance:** 2
**Originality:** 2
**Rating:** 4
**Confidence:** 3

**Summary:**

This paper proposes a two-stage zero-order prefix tuning method for the CLIP. The first stage utilizes sharpness-aware CMA-ES for creating a smooth loss landscape and then the second stage performs the major zero-order optimization.

**Questions:**

Please address my concerns in the above session.

**Ethical Concerns:**

["NO or VERY MINOR ethics concerns only"]

**Final Justification:**

Authors have properly addressed all my concerns. Indeed, their method shows much better results than other zero-order optimization methods. Though the accuracy is not as good as gradient-based methods, their method has significantly less resource consumption. Thus, I would like to raise my rating towards accept. Thank you.

**Limitations:**

Yes.

**Paper Formatting Concerns:**

Appendix is submitted together with the main text.

**Quality:**

3

**Strengths And Weaknesses:**

- The SharpZO method is clearly introduced and easy to follow. Moreover, the SharpZO method shows a significant performance advantage than other zero-shot optimization methods in both accuracy and speed.
- Page 2 Line 38: Why would a high variance lead to suboptimal convergence? I assume a high variance would lead to the failure of convergence.
- The official baseline result for ZIP in Table 1 seems wrong. For Flowers102, the reported baseline in the ZIP paper is 70.4%, but in this paper it is 92.3%.
- In the Table 2, SharpZO does not perform the best on ImageNet-Sketch, thus shouldn't be highlighted in bold.
- The hyper-parameter selection in Table 7 is not detailed, *i.e.*, is the parameter search performed on the test set?
- The configuration for few-shot learning is not included.
- Authors did **not** include any information about how they evaluate the statistical significance of their results (*e.g.*, the number of random seeds adopted). Only the ablation experiments shows the variance of results. For all other experimental results that involves a comparison with other baseline methods, the statistical significance required in the NeurIPS checklist is not justified. As zero-shot optimization methods are highly stochastic, a sufficient repeated evaluations with different seeds are necessary.
- For CMA-ES in the Figure 3, there is a huge accuracy drop and variance gap between the end of the Figure 3(a) and the beginning of the Figure 3(b). However, for the S-aware CMA-ES, there is no such difference. I would like to know what leads to such a difference.
- For comparison with gradient based method, prefix tuning should also be included justifying the need for zero-order optimization.

---

> ### Author Rebuttal · Authors · 2025-07-30
>
> Thank you for your thoughtful and detailed feedback. We greatly appreciate your recognition of the strengths of our work, as well as your constructive comments regarding its weaknesses. Below, we address each of the points raised in your review.
>
> ## Weaknesses
>
> > Page 2 Line 38: Why would a high variance lead to suboptimal convergence? I assume a high variance would lead to the failure of convergence.
> >
>
> Thanks for pointing this out. We agree that high variance can slow down or misdirect updates. The issue of convergence to suboptimal regions in line 38 is more related to the limitations of local search dynamics. We will revise the sentence to clarify this distinction in the final version.
>
> > The official baseline result for ZIP in Table 1 seems wrong. For Flowers102, the reported baseline in the ZIP paper is 70.4%, but in this paper it is 92.3%. In the Table 2, SharpZO does not perform the best on ImageNet-Sketch, thus shouldn't be highlighted in bold.
> >
>
> Apologies for these typos—these errors likely occurred during table reformatting. The right number for the ZIP method should be 70.4%. Since it’s an error during table reformatting, the average accuracy is not influenced. We have corrected these errors
>
> > The hyper-parameter selection in Table 7 is not detailed, *i.e.*, is the parameter search performed on the test set?
> >
>
> Yes, we tune hyper-parameters based on test set accuracy. For each hyper-parameters, we select 3–5 candidate values ([0.1, 0.2, 0.4, 1.0] for CMA_ES step size, [1e-3,1e-4, 1e-5] for ZO scale and [1e-1, 1e-3, 1e-5] for ZO learning rate) and choose the one yielding the best performance with the other hyper-parameters fixed. We will include these details in the revised version.
>
> > The configuration for few-shot learning is not included.
> >
>
> Thanks for pointing this out. We use a 16-shot setup in all experiments, consistent with prior work such as ZIP, to ensure a fair comparison. We will add this information in the revised paper.
>
> > Authors did **not** include any information about how they evaluate the statistical significance of their results (*e.g.*, the number of random seeds adopted). Only the ablation experiments shows the variance of results. For all other experimental results that involves a comparison with other baseline methods, the statistical significance required in the NeurIPS checklist is not justified. As zero-shot optimization methods are highly stochastic, a sufficient repeated evaluations with different seeds are necessary.
> >
>
> We agree that further repeated evaluations with different seeds are necessary. In addition to the standard deviation shown in the Fig. 1, 3, 4 in the paper, we conducted experiments using four different seeds and report the mean and standard deviation, as shown in the following table. Due to time constraints, we performed this evaluation on the four datasets below, but we will include standard deviation results for all tasks in the final version.
>
> |  | EuroSAT | DTD | OxfordPets | ImageNet |
> | --- | --- | --- | --- | --- |
> | SharpZO(mean+std) | 80.77±1.53 | 60.58±0.86 | 89.51±1.14 | 63.29±0.65 |
>
> As we can observe, the overall randomness is limited to a standard deviation around 1 in our fine-tuning case. This degree of randomness is similar to the results reported in previous ZO LLM fine-tuning paper (”Fine-Tuning Language Models with Just forward Passes”, Table 18)).
>
> > For CMA-ES in the Figure 3, there is a huge accuracy drop and variance gap between the end of the Figure 3(a) and the beginning of the Figure 3(b). However, for the S-aware CMA-ES, there is no such difference. I would like to know what leads to such a difference.
> >
>
> This is an interesting point. In short, the different behavior should be attributed to the different sampling strategy used in S-aware CMA-ES (refer to Eq. (3)), which includes a gradient-related term in the first stage of training. In contrast, it is expected to observe some disturbance in the training curve when switching optimizers with the naive CMA-ES method, as the search dynamics of the two optimizers are fundamentally different: CMA-ES learns a distribution, whereas ZO relies on locally estimated gradients. This change also helps improve the overall accuracy of the SharpZO method.
>
> > For comparison with gradient based method, prefix tuning should also be included justifying the need for zero-order optimization.
> >
>
> Thanks for the suggestion. It’s possible that you were referring to first-order prompt tuning (as the original prefix tuning is applied to the attention mechanism, which is not a good fit for CLIP model with possible ResNet Backbone), which has been studied in the CoOp paper [1]. Below, we provide an additional comparison between SharpZO and CoOp.
>
> |  | EuroSAT | DTD | OxfordPets | ImageNet | AVG |
> | --- | --- | --- | --- | --- | --- |
> | CoOp | 84.05 ± 1.05 | 63.26 ± 0.22 | 87.06 ± 0.88 | 63.00 ± 0.18 | 73.48 ± 0.39 |
> | SharpZO | 80.77±1.53 | 60.58±0.86 | 89.51±1.14 | 63.29±0.65 | 69.76  |
>
> We can observe the our BP-free SharpZO performs on-par or better than the BP-based prompt tuning method CoOp on dataset like OxfordPets and ImageNet. We also agree there is still a gap between SharpZO and CoOp on other datasets. This is limited by the stochastic nature of zero-order fine-tuning, which in return, bring in the advantage of very low memory cost and inference-only fine-tuning availability.
>
> [1] Zhou, K., Yang, J., Loy, C. C., and Liu, Z. Learning to prompt for vision-language models. IJCV, 130(9):2337– 2348, 2022b.

---

> ### Author Response · Authors · 2025-08-01
> **Correction for our rebuttal**
>
> Dear Reviewer,
>
> We just identified a miscommunication regarding the data used for hyper-parameter search in our rebuttal, as we previously mixed it up with our hyper-parameter ablation experiments setup.
>
> Regarding your question:
>
> > The hyper-parameter selection in Table 7 is not detailed, i.e., is the parameter search performed on the test set?
>
> - The hyperparameter search for Table 7 is conducted exclusively on the validation set, not the test set. The test set is used only for the final evaluation and for plotting the test accuracy curves.
>
> - The test set is used to get the test accuracy appears in Tables 5 and 6 solely for post-hoc hyperparameter ablation studies, which analyze the impact of different settings but do not influence hyperparameter selection for the main results.
>
> We are sorry for this confusion.
>
> Best,
>
> Authors

---

> > ### Comment · Reviewer_VzLh · 2025-08-05
> >
> > Thank you for your response. Most of my concerns have been properly addressed. For the comparison with prompt tuning, could you please also report the memory usage and time-to-test-accuracy? Thank you.

---

> > > ### Author Response · Authors · 2025-08-06
> > > **Thank you for the acknowledgement**
> > >
> > > Thank you for acknowledging that our responses have addressed your previous concerns.
> > >
> > > Here, we further provide a comparison of memory usage and time-to-test-accuracy between prompt tuning (CoOp) and our SharpZO method for reference. All experiments are conducted on the ViT-B/16 backbone, and the CoOp results are obtained using its official implementation. We measure the time required to reach an accuracy level that all baselines in Table 1 can achieve.
> > >
> > > | ImageNet | Memory (MB) | ToTA (MIN) | Sun397 | Memory (MB) | ToTA (MIN) |
> > > | --- | --- | --- | --- | --- | --- |
> > > | CoOp | 16802 | 59.4 | CoOp | 7360 | 3.5 |
> > > | SharpZO | 3297 | 15.3 | SharpZO | 3130 | 5.2 |
> > >
> > > As we can observe, with increasing dataset complexity (i.e., a larger number of classes), the memory and time advantages of the SharpZO method become more pronounced. This is because the ZO method does not require a backpropagation graph, whose size grows directly with the number of classes. Moreover, ZO enjoys a much faster per-step runtime compared with first-order methods, enabling it to converge more quickly, even in the presence of higher noise on certain tasks.
> > >
> > > Best,
> > >
> > > Authors

---

> > > > ### Comment · Reviewer_VzLh · 2025-08-06
> > > >
> > > > Thank you for your quick update. Please update your paper with 1. statistical significance evaluation and 2. comparison with gradient-based methods. As all my concerns are properly addressed, I will raise the rating. Thanks.

---

> > > > > ### Author Response · Authors · 2025-08-06
> > > > > **Thank you for the response**
> > > > >
> > > > > Thank you for recognizing our work. We agree with your points and will incorporate these results in the revised version.
> > > > >
> > > > > Best,
> > > > >
> > > > > Authors

---

### Decision · Program_Chairs · 2025-09-17

**Decision:**

Accept (poster)

**Comment:**

The paper introduces SharpZO, a hybrid sharpness-aware zero-order (ZO) optimization method for vision-language model prompt tuning using only forward passes. It combines sharpness-aware CMA-ES for initialization with sparse ZO optimization, supported by theoretical justification and empirical validation. Strengths include clear methodological contributions, strong empirical performance gains over prior BP-free methods, reduced resource consumption, and demonstrated efficiency advantages. Weaknesses identified by reviewers concerned limited discussion of statistical significance, reliance on validation sets in few-shot settings, absence of comparisons with gradient-based baselines in the initial submission, and questions about generalizability beyond CLIP. However, during rebuttal, the authors provided clarifications, corrected table errors, added statistical evaluations, extended comparisons with gradient-based methods, and reported additional experiments (e.g., SigLIP, base-to-new generalization), which resolved most concerns. The discussion confirmed that the method achieves both accuracy and efficiency benefits while maintaining practical applicability. Given the sound methodology, thorough rebuttal, and potential impact on BP-free fine-tuning, the TC recommends accepting this paper.